# Multiple ParA/MinD ATPases coordinate the positioning of disparate cargos in a bacterial cell

Lisa T. Pulianmackal [1], Jose Miguel I. Limcaoco[2], Keerthikka Ravi [3], Sinyu Yang[3], Jeffrey Zhang[3], Mimi K. Tran[3], Maria Ghalmi[3], Matthew J. O'Meara [2] & Anthony G. Vecchiarelli [3] ✉

In eukaryotes, linear motor proteins govern intracellular transport and organization. In bacteria, where linear motors involved in spatial regulation are absent, the ParA/MinD family of ATPases organize an array of genetic- and protein-based cellular cargos. The positioning of these cargos has been independently investigated to varying degrees in several bacterial species. However, it remains unclear how multiple ParA/MinD ATPases can coordinate the positioning of diverse cargos in the same cell. Here, we find that over a third of sequenced bacterial genomes encode multiple ParA/MinD ATPases. We identify an organism (*Halothiobacillus neapolitanus*) with seven ParA/MinD ATPases, demonstrate that five of these are each dedicated to the spatial regulation of a single cellular cargo, and define potential specificity determinants for each system. Furthermore, we show how these positioning reactions can influence each other, stressing the importance of understanding how organelle trafficking, chromosome segregation, and cell division are coordinated in bacterial cells. Together, our data show how multiple ParA/MinD ATPases coexist and function to position a diverse set of fundamental cargos in the same bacterial cell.

Actin filaments, microtubules, and the linear motor proteins that walk along them, are well known for spatial organization in eukaryotic cells. In bacteria, however, where linear motors involved in positioning are absent, a widespread family of ParA/MinD (A/D) ATPases spatially organize plasmids, chromosomes, and an array of protein-based organelles, many of which are fundamental to cell survival and pathogenesis. By far the two best studied ATPases, and family namesake, are ParA involved in plasmid partition and chromosome segregation[1,2], and MinD involved in divisome positioning[3]. Less studied is the growing list of A/D ATPases, widespread across prokaryotes, involved in spatially regulating diverse protein-based organelles, such as Bacterial Microcompartments (BMCs)[4,5], flagella[6,7], chemotaxis clusters[8,9], and conjugation machinery[10].

Despite the cargos being so diverse, A/D ATPases share a number of features: (i) all form ATP-sandwich dimers[11], (ii) dimerization forms an interface for binding a positioning matrix—the nucleoid for ParA-like ATPases[12,13] or the inner membrane for MinD-like ATPases[14,15], and (iii) dimerization also forms a binding site for a cognate partner protein that connects an ATPase to its cargo and stimulates its release from the positioning matrix. For example, in chromosome segregation, the ParA partner is ParB, which loads onto a centromere-like site, called *parS*, to form a massive complex on the chromosome near the origin of replication (*OriC*)[2]. This ParB-*parS* complex locally stimulates ParA ATPase activity and nucleoid release, which generates ParA gradients on the nucleoid. Segregation of sister chromosomes ensues as the ParB-*parS* complex chases

[1]Department of Microbiology and Immunology, University of Michigan, Ann Arbor, MI 48109, USA. [2]Department of Computational Medicine & Bioinformatics, University of Michigan, Ann Arbor, MI 48109, USA. [3]Department of Molecular, Cellular, and Developmental Biology, University of Michigan, Ann Arbor, MI 48109, USA. ✉e-mail: ave@umich.edu

nucleoid-bound ParA gradients in opposite directions[16]. Therefore, unlike the mitotic-spindle apparatus used in eukaryotic chromosome segregation, prokaryotes use a fundamentally different mode of spatial organization—A/D ATPases make waves on biological surfaces to position their respective cargos.

Chromosome segregation, cell division positioning, and organelle trafficking reactions have been independently investigated to varying degrees in several prokaryotes. Yet, it remains unknown how many A/D ATPases can be encoded in a single bacterium to position multiple disparate cargos, or how bacteria spatiotemporally coordinate the positioning of such a diverse set of fundamental cargos in the same cell. Additionally, the mechanistic variations and specificity determinants that govern the positioning of such a diverse set of cellular cargo remain unclear. This is because A/D-based positioning reactions are typically studied independently of one another and in model bacteria with few A/D ATPases.

Here, we find that a third of sequenced bacteria encode multiple A/D ATPases. Among these bacteria, we identified *Halothiobacillus neapolitanus* (*H. neapolitanus* hereafter), with seven putative A/D ATPases. Neighborhood analysis of the A/D ATPase genes in *H. neapolitanus* implicate several putative cargos. We use genetics and cell biology to assign five of the A/D ATPases in *H. neapolitanus* to their cargos. Our findings show that each ATPase is directly dedicated to the positioning and faithful inheritance of a specific cargo type. The tractability and number of A/D ATPases made *H. neapolitanus* a valuable tool for also investigating how bacteria coordinate the processes of chromosome segregation and cell division with organelle trafficking—a well-studied question in eukaryotic cells that remains unaddressed in prokaryotes. We show how the deletion of one A/D ATPase can have indirect effects on the inheritance of disparate cargos positioned by other A/D ATPases via defects in DNA replication, chromosome segregation, and/or cell division. We also provide evidence that flagella positioning influences the spatial regulation of chemotaxis clusters. Finally, we identify putative sequence- and structural-determinants that uniquely link each A/D ATPase to a specific cargo, ultimately allowing these related ATPases to coexist and function in the same cell. Together, our study probes mechanistic commonality and variation in the most widespread ATPase family used in the spatial regulation of diverse cellular cargos across prokaryotes, and all within a single cell.

## Results

### A third of sequenced bacteria encode multiple ParA/MinD family ATPases

It is unclear how many ParA/MinD (A/D) family ATPases are encoded in a single organism. To answer this question, we performed an extensive tBLASTn analysis using a consensus protein sequence, generated from well-studied A/D ATPases, as the query (see methods). As already established[17], we found that ~ 96% of bacteria from the NCBI Reference Sequence (RefSeq) database encode at least one A/D ATPase (Supplementary Data 1 and 2). These hits were binned by bacterial species, which were then ordered by the number of A/D hits. From this initial list, we found many bacterial genomes encoding 10 to 20 A/D ATPases. However, these bacteria with the most A/D ATPases had their genomes encoded on multiple plasmids and chromosomes, each of which encode its own ParA-based DNA segregation system[18]. For this study, we were specifically focused on understanding how multiple A/D ATPases coexist and coordinate the positioning of *disparate* cargos in the same cell. Therefore, we further filtered our dataset to identify bacteria encoding multiple A/D ATPases, but only one chromosome and no stable plasmids (Supplementary Data 1 and 2). Even after accounting for bacteria with genomes encoded on multiple genetic elements, our bioinformatic analysis revealed that more than a third of sequenced bacteria encode multiple A/D ATPases (Fig. 1a, Supplementary Data 1 and 2).

### Gene neighborhood analysis of A/D ATPases in *H. neapolitanus* implicate cellular cargos

We next set out to determine how multiple A/D ATPases can coexist in the same cell to position disparate cargos. To address this question, we identified an organism from our list of bacteria encoding multiple A/D ATPases (Supplementary Data 1 and 2). Among the top 1% of bacteria (encoding six or more A/D ATPases), we identified several pathogens including *Clostridia*, *Burkholderia*, *Mycobacteria*, *Vibrio*, *Pseudomonas*, and *Xanthomonas* species. We also identified the non-pathogenic and experimentally tractable organism, *H. neapolitanus*—a slow-growing, sulfur-oxidizing chemoautotroph that encodes seven putative A/D ATPases on one chromosome (Fig. 1b). Spatial regulation by A/D ATPases has largely been studied in fast-growing model bacteria. We intentionally chose a slow growing bacterium (6 h doubling time) because the infrequent DNA segregation and cell division events allowed for larger observation windows and a more direct view into the dynamics of organelle trafficking. The tractability, slow-growth rate, and abundance of A/D ATPases made *H. neapolitanus* an ideal choice for further study.

To determine whether the A/D ATPase hits in *H. neapolitanus* were indeed spatial regulators, we performed a gene neighborhood analysis (GNA) to identify putative cargos (Fig. 1b). GNA allows us to infer function because A/D ATPase genes are often encoded near cargo-associated loci. For example, the ParAB*S* chromosome segregation system is typically encoded near *OriC*[19]. Strikingly, the putative cargos we identified using this approach includes the chromosome and all known protein-based cargos of the A/D ATPase family[20,21]. While spatial regulation of these diverse cargos has been individually studied in many different model bacteria, their coordinated positioning by multiple A/D ATPases has never been investigated together in one organism.

As a second line of bioinformatic evidence linking each A/D ATPase to a specific cargo, we investigated the conservation of the A/D ATPase gene neighborhoods using FlaGs (Flanking Genes) analysis[22]. FlaGs analysis predicts functional associations by taking a list of NCBI protein accessions as input and clusters neighborhood-encoded proteins into homologous groups. Homologs of each A/D ATPase in *H. neapolitanus* were identified using BLASTp, and top hits were used as input for FlaGs analysis (Supplementary Fig. 1). The analysis shows strong conservation of the A/D ATPase gene neighborhoods across multiple bacterial phyla, further implicating the putative cargos. Due to the limited data on A/D ATPases associated with conjugation[10] or nitrogen metabolism, we excluded these hits from further investigation in this study.

With the remaining five A/D ATPases, we performed a multiple sequence alignment against ParA/MinD family members that have been previously established to position a cellular cargo (Fig. 1c). Each A/D ATPase in *H. neapolitanus* clustered with a specific family member known to position a specific cellular cargo in other bacteria—the chromosome (ParA[2]), the divisome (MinD[23]), the carboxysome (McdA[4,5]), the flagellum (FlhG[6,7]) and the chemotaxis cluster (ParC[8,9]). These data provide a third line of evidence that further implicates the putative cargos identified by our GNA. We next sought to directly identify the role of each A/D ATPase in positioning the cargos implicated by bioinformatics.

### The ParAB system is required for chromosome segregation in *H. neapolitanus*

Chromosome segregation prior to cell division is critical for cellular survival. In most bacteria, faithful chromosome segregation and inheritance are mediated by a ParAB system encoded near *OriC*[19]. Without ParAB, DNA is asymmetrically inherited, resulting in anucleate and polyploid cells, and reduced cell fitness or death[2]. In *H. neapolitanus*, there is a putative *parAB* system encoded near *OriC* (Fig. 2a). To image chromosome segregation, the ParB homolog, encoded downstream of

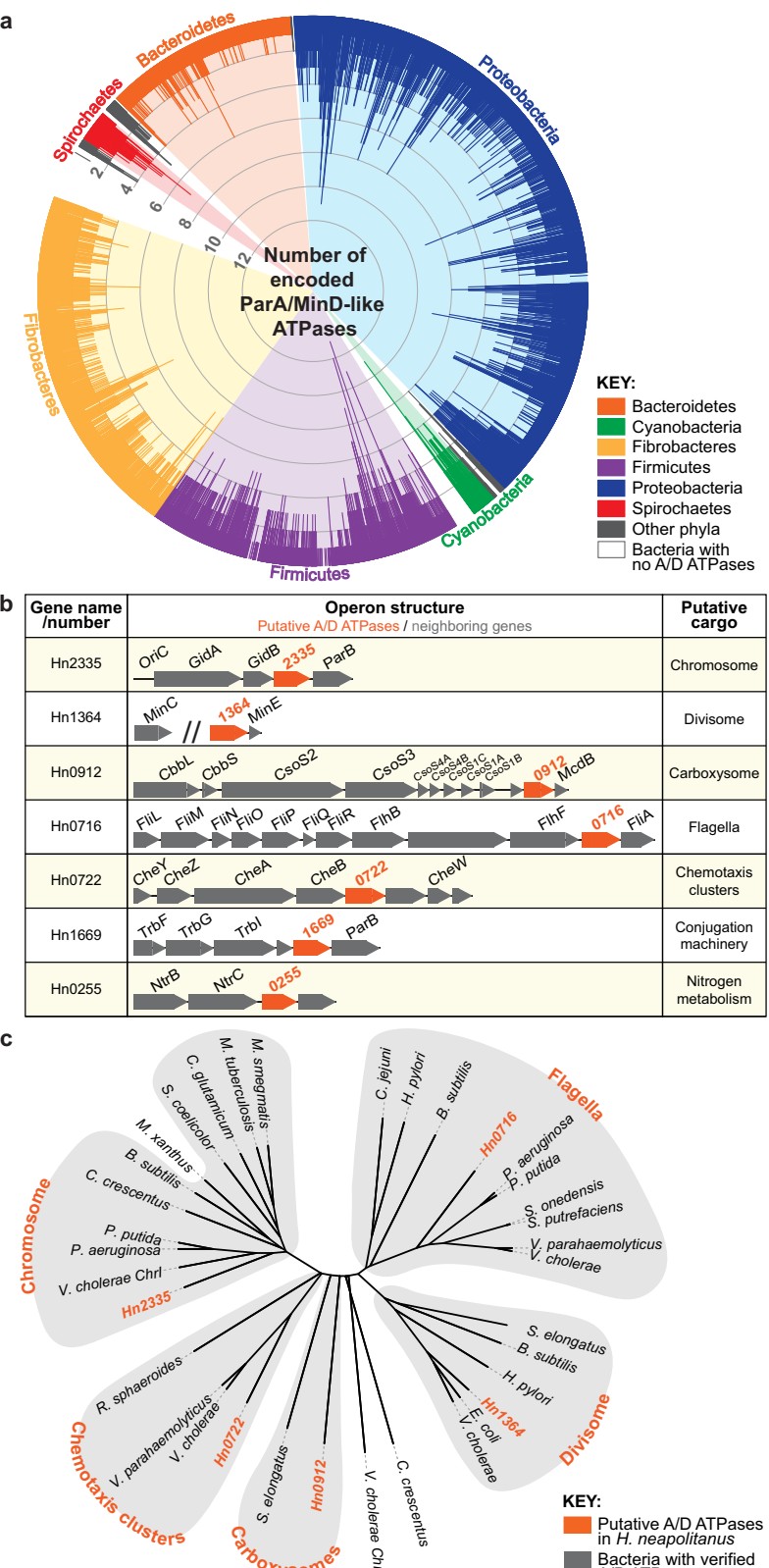

**Fig. 1 | ParA/MinD-like ATPases are widespread in bacteria. a** 96% of sequenced bacterial genomes encode at least one A/D ATPase and 35% encode multiple. Each spike represents a bacterial species and spike length indicates the number of unique A/D hits per bacterium. **b** *H. neapolitanus* encodes seven putative ParA/MinD-like positioning systems. Gene neighborhood analysis implicates the putative cargos associated with each putative A/D ATPase. **c** Multiple sequence alignment of each A/D ATPase from *H. neapolitanus* against experimentally-verified ParA/MinD family members further implicates the putative cargos identified by gene neighborhood analysis. Each of the putative A/D ATPases in *H. neapolitanus* cluster with family members shown to position the indicated cellular cargos (orange) in other bacteria.

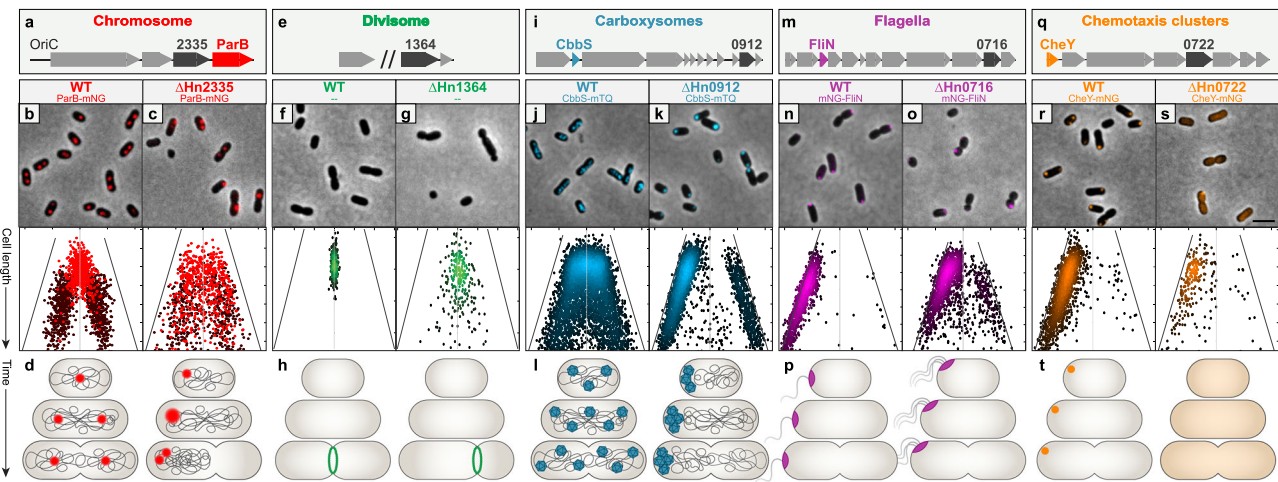

**Fig. 2 | Each A/D ATPase in *H. neapolitanus* positions a specific cargo.**
**a–d** *Hn2335* is required for chromosome segregation. **a** *Hn2335* is found near *OriC* and has a ParB-homolog encoded downstream. **b, c** The origin region of the chromosome was tagged by labelling the ParB homolog. Light red: 1 focus/cell; dark red: 2 foci/cell. Short WT cells had a single focus at mid-cell, whereas longer cells had two foci at the quarter positions. Δ*Hn2335* cells displayed random positioning of ParB foci regardless of cell length. **d** Cartoon diagrams depict chromosome segregation in WT and Δ*Hn2335* cells. **e–h** *Hn1364* is required for divisome positioning. **e** *Hn1364* is found upstream of *minE*. **f, g** Divisome positioning was determined by the location of constriction sites. Each dot on the density graph represents one identified constriction site. In WT cells, constriction sites were at mid-cell. In Δ*Hn1364*, constriction sites were found across the cell length. **h** Cartoon diagrams depict cell division in WT and Δ*Hn1364* cells. **i–l** Carboxysome positioning is determined by McdA. **i** *HnO912 (mcdA)* is found near genes encoding carboxysome shell proteins and Rubisco. **j, k** Carboxysomes were visualized by labelling the small subunit of the Rubisco enzyme, *cbbS*. Each dot on the density graph represents one carboxysome focus. In WT cells, carboxysomes are distributed along the cell length. In Δ*mcdA*, carboxysomes formed large polar foci at one or both poles. **l** Cartoon diagrams depict carboxysome distribution in WT and Δ*HnO912* cells. **m–p** *HnO716* is required for regulating flagella position and copy number. **m** *HnO716* is found near flagella-associated genes. **n, o** Flagella localization was visualized by labelling a component of the flagellar basal body, *fliN*. Each dot on the density graph represents one FliN focus. WT cells had a single polar FliN focus. In

Δ*HnO716* cells, FliN foci were more randomly positioned along the cell length. **p** Cartoon diagrams depict flagella localization and number in WT and Δ*HnO716* cells. **q–t** *HnO722* is required for chemotaxis cluster positioning. **q** *HnO722* is found near chemotaxis-associated genes. **r, s** Chemotaxis clusters were visualized by labelling the response regulator, *cheY*. Each dot on the density graph represents one CheY focus. WT cells had a single CheY polar focus. Δ*HnO722* mutant cells typically had no CheY foci. **t** Cartoon diagrams depict chemotaxis clusters in WT and Δ*HnO722* cells. (All images) Micrographs shown are representative images from at least 3 experiments. Scale bar: 2 μm. (All graphs) Cells were analyzed and quantified using MicrobeJ. On the *y*-axis, cells are sorted by increasing cell length, with shorter cells at the top and longer cells at the bottom. The *x*-axis represents the distance from mid-cell in microns; the center vertical line equates to a distance of zero from mid-cell. Each dot represents where a focus was found along the length of the cell. For all density plots, lighter colors represent higher density, darker colors represent lower density. For flagella, chemotaxis, and carboxysome graphs, the cell pole closest to a focus was oriented to the left; foci in the right half of the graph indicate the presence of a second focus. Graph axes for chromosome, carboxysome, flagella, and chemotaxis-labelled mutants: *y*-axis range (cell length): 0.8–2.1 μm; *x*-axis range (distance from mid-cell): −1.1–1.1 μm. *X*-axis of Δ*flhG* cells is 0.5–1.8 μm. Graph axes for divisome: *y*-axis range (cell length): 1.5–3.0 μm; *x*-axis range (distance from mid-cell): −1.5–1.5 μm. Source data are provided as a Source Data file.

the putative *parA* gene (*Hn2335*), was fused to the fluorescent protein monomeric Neon Green (mNG). ParB-mNG was observed as one or two puncta per cell (Fig. 2b). Population analysis showed that shorter cells had a single focus at mid-cell, whereas longer cells had two foci at the quarter positions of the cell (Fig. 2b). When the putative *parA* gene (*Hn2335*) was deleted, ParB-mNG foci were completely absent in 25% of cells (Supplementary Fig. 2d). When ParB foci were present, they were randomly positioned regardless of cell length (Fig. 2c) and significantly brighter compared to that of WT (wild-type) cells (Supplementary Fig. 2e). Complementation with *Hn2335* at an exogenous locus restored foci positioning (Supplementary Fig. 2h). These data suggest that replicated chromosomes in Δ*Hn2335* were no longer faithfully segregated, resulting in anucleate and polyploid cells.

We then performed time-lapse microscopy of ParB-mNG foci and SYTOX-stained nucleoids to observe chromosome segregation in real time. Newborn WT cells have a single ParB focus at mid-cell, which then splits into two foci that bidirectionally segregate towards the quarter positions of the growing cell (Supplementary Fig. 2f and Supplementary Movie 1A). Foci positioning at the cell quarters was maintained, which then became the mid-cell position of each daughter cell following division. Faithful chromosome segregation and inheritance were lost when the putative *parA* gene (*Hn2335*) was deleted (Supplementary Fig. 2g and Supplementary Movie 1B). Many cells had a single large ParB focus at a cell pole that did not split and all DNA was

concentrated into a single daughter cell upon division. These polyploid daughters continued to divide, while the anucleate daughters were no longer viable. These data confirmed that the increased ParB foci intensities in the deletion mutant represents an increase in chromosome copy number in these cells. In summary, the A/D ATPase encoded by *Hn2335* is required for faithful chromosome segregation (Fig. 2d) and we will henceforth refer to this protein as ParA.

### The MinCDE system aligns cell division at mid-cell in *H. neapolitanus*

Proper positioning of the divisome ensures that when a cell divides, both daughter cells are roughly equal in length. Without the Min system, division occurs at any nucleoid-free region[24], producing anucleate minicells, the products of polar divisions. Our bioinformatic analyses showed that the protein encoded by *Hn1364* is a MinD homolog, and immediately downstream of this gene in the same operon is a gene encoding a MinE homolog (Fig. 2e). To determine if *Hn1364* was indeed involved in divisome positioning, we analyzed dividing cells and identified their constriction sites relative to cell length (Fig. 2f, g). WT cells divided close to mid-cell (Fig. 2f and Supplementary Fig. 3d) while the Δ*Hn1364* cells divided asymmetrically (Fig. 2g and Supplementary Fig. 3e). Population analysis of dividing cells identified mid-cell constriction sites in 97% of WT cells compared to only 41% of Δ*Hn1364* cells (Supplementary Fig. 3f and Supplementary Movie 2). In

addition to single asymmetric division events, 9% of dividing cells in the Δ*Hn1364* mutant population formed multiple division sites simultaneously along the cell length (Supplementary Fig. 3e, Supplementary Fig. 3g and Supplementary Movie 2C). The unequal divisions resulted in greater variation in cell length (Supplementary Fig. 3h). Complementation with *Hn1364* at an exogenous locus restored mid-cell constriction and single division events (Supplementary Fig. 3j). Overall, our findings show that *Hn1364* is critical for positioning the divisome at mid-cell (Fig. 2h) and we will henceforth refer to this protein as MinD.

## The A/D ATPase, McdA, encoded in the carboxysome operon positions carboxysomes

Bacterial microcompartments, or BMCs, are large icosahedral protein-based organelles that encapsulate sensitive metabolic reactions to provide prokaryotes with distinct growth advantages[25]. Despite their importance, little is known about how BMCs are spatially regulated. The model BMC is the carbon-fixing carboxysome found in cyanobacteria and proteobacteria[26]. It was recently found that an A/D ATPase, we termed <u>M</u>aintenance of <u>c</u>arboxysome <u>d</u>istribution protein <u>A</u> (McdA), is widespread among carboxysome-containing bacteria, including *H. neapolitanus* (Fig. 2i)[4]. McdA spaces carboxysomes on the nucleoid along with its partner protein, McdB[27,28]. To demonstrate its requirement for carboxysome positioning, we visualized carboxysomes by labelling the small subunit of the encapsulated Rubisco enzyme (*cbbS*) with mTurquoise2 to form CbbS-mTQ. As previously shown, in WT cells, carboxysomes are distributed down the cell length (Fig. 2j, Supplementary Fig. 4d). In the deletion mutant (Δ*mcdA*), carboxysome aggregates form a large bright polar focus at one or occasionally both poles (Fig. 2k, Supplementary Fig. 4d, e). Complementation with *mcdAB* at an exogenous locus restored carboxysome positioning (Supplementary Fig 4h). These data are consistent with our previous observations using TEM, which showed that foci in the mutant population represent an aggregation of assembled carboxysomes[28].

We extend our previous findings here with long-term time-lapse microscopy. In WT cells, carboxysomes are dynamically positioned along the cell length across multiple generations (Supplementary Fig. 4f and Supplementary Movie 3A). In the deletion mutant, polar carboxysome aggregates were stagnant (Supplementary Fig. 4g and Supplementary Movie 3B). Together, our findings show that the A/D ATPase encoded within the carboxysome operon, we termed McdA, is essential for distributing carboxysomes across the cell length and ensuring organelle homeostasis following division (Fig. 2l).

## *Hn0716* is required for regulating flagella number and positioning

Flagella are external filamentous structures that allow for bacterial motility. Bacteria vary in flagella location, number, and pattern. Many bacteria encode an A/D ATPase called FlhG in their flagellar operon, which is essential for diverse flagellation patterns in many bacteria[7], yet the mechanisms remain unclear (Supplementary Discussion). In polar flagellates, deletion of *flhG* typically results in changes in flagella number and motility[6,29–34]. Meanwhile, in the peritrichous organism, *Bacillus subtilis*, deletion of *flhG* results in changes to flagella location[35]. Surprisingly, deletion of *flhG* in the amphitrichous organism, *Campylobacter jejuni*, also results in cell division defects[36,37].

Our bioinformatic analysis suggests that *Hn0716* within the flagella operon encodes an FlhG homolog (see Fig. 1c). To determine if *Hn0716* is involved in flagellar spatial regulation, we first visualized a mNG fusion of FliN, which is a component of the flagellar basal body that assembles at the cytoplasmic face of the membrane (Fig. 2m)[38]. WT cells had a single mNG-FliN focus at the extreme cell pole (Fig. 2n). In Δ*Hn0716* cells, FliN foci were no longer faithfully positioned (Fig. 2o) and cells were more likely to have zero or multiple foci (Supplementary Fig. 5d). Complementation with *Hn0716* at an exogenous locus

restored FliN foci localization to the poles (Supplementary Fig. 5i). These data suggest that *Hn0716* is required for positioning a single flagellum at a single pole in *H. neapolitanus*.

We next set out to determine whether these alterations to FliN positioning affected cell motility. Motility assays in soft agar found that Δ*Hn0716* cells were not motile (Supplementary Fig. 5g). Loss of motility could be due to flagellar mispositioning, a loss of flagella, or hyper-flagellation. To image flagella, we engineered flagellin[T185C], which allows for fluorescent labeling of flagella via the addition of a cysteine-reactive maleimide stain to the media[39]. We found that WT *H. neapolitanus* cells are monotrichous, with a single polar flagellum emanating from the FliN focus (Supplementary.Fig. 5h). Δ*Hn0716* cells also had flagella emanating from FliN foci, however, the cells were hyper-flagellated, with multiple flagella often bundled together as tufts, emanating from several FliN foci. We conclude that *Hn0716* is required for regulating flagella number and position in *H. neapolitanus* (Fig. 2p) and we will henceforth refer to this protein as FlhG. Future studies will investigate the unique pleiotropic effects of *flhG* in flagella assembly, number, and location, as well as cell division in *H. neapolitanus* (Supplementary Discussion).

## *Hn0722* is required for chemotaxis cluster assembly and positioning

Directing bacterial motility are large hexagonal arrays called chemotaxis clusters, comprised of chemoreceptors, an adaptor protein (CheW), and a kinase (CheA). Several mechanisms have evolved to control both the number and positioning of chemotaxis clusters, including the use of A/D ATPases (called ParC in *Vibrio* species or PpfA in *R. sphaeroides*). In *Vibrio* species, ParC directs chemotaxis arrays to a polar landmark protein called HubP[8,40]. As a consequence, daughter cells inherit an array at their old pole upon cell division. In *R. sphaeroides*, there are no polar landmarks and PpfA distributes chemotaxis clusters over the nucleoid[9,41]. Where studied, deletion of the A/D ATPase alters chemotaxis cluster number and positioning in cells, which results in a reduction in spreading motility in soft agar.

Our bioinformatics analysis showed that the protein encoded by *Hn0722* is a ParC/PpfA homolog within the chemotaxis operon of *H. neapolitanus* (see Fig. 1c). To determine if *Hn0722* is important for spatially regulating chemotaxis clusters, we first imaged chemotaxis clusters by fusing mNG to CheY (Fig. 2q). CheY is a response regulator that is phosphorylated by CheA, and has previously been shown to colocalize with chemotaxis clusters in *E. coli*[42,43]. Consistent with electron micrographs of chemotaxis clusters in *H. neapolitanus*[44], CheY-mNG formed a single focus immediately proximal to one cell pole in ~ 85% of WT cells (Fig. 2r, Supplementary Fig. 6d). However, when *Hn0722* was deleted, the CheY-mNG signal was diffuse in ~ 80% of the cell population (Fig. 2s, Supplementary Fig. 6d). In the ~20% of Δ*Hn0722* cells that had a CheY-mNG focus, the foci were significantly lower in intensity (Supplementary Fig. 6e). Complementation with the overlapping genes *Hn0722* and *Hn0723* recovered CheY foci at the poles (Supplementary Fig. 6f). Together, we conclude that *Hn0722* is required for chemotaxis cluster assembly and positioning in *H. neapolitanus* (Fig. 2t) and we will henceforth refer to this protein as ParC.

## Cargo positioning is not directly controlled by A/D ATPases encoded at distant loci

We have thus far provided direct evidence showing that five A/D ATPases position five disparate cellular cargos in *H. neapolitanus* (Fig. 2). We next asked if each of the five positioning reactions occurred independently from each other. To answer this question, we individually deleted each A/D ATPase in every cargo-labeled background strain (Fig. 3). We found carboxysome (Fig. 3c) and flagella (Fig. 3d) positioning were largely unaffected by the deletion of distant A/D ATPases. Intriguingly, chromosome (Fig. 3a), divisome (Fig. 3b), and chemotaxis cluster (Fig. 3e) positioning or focus intensity were all

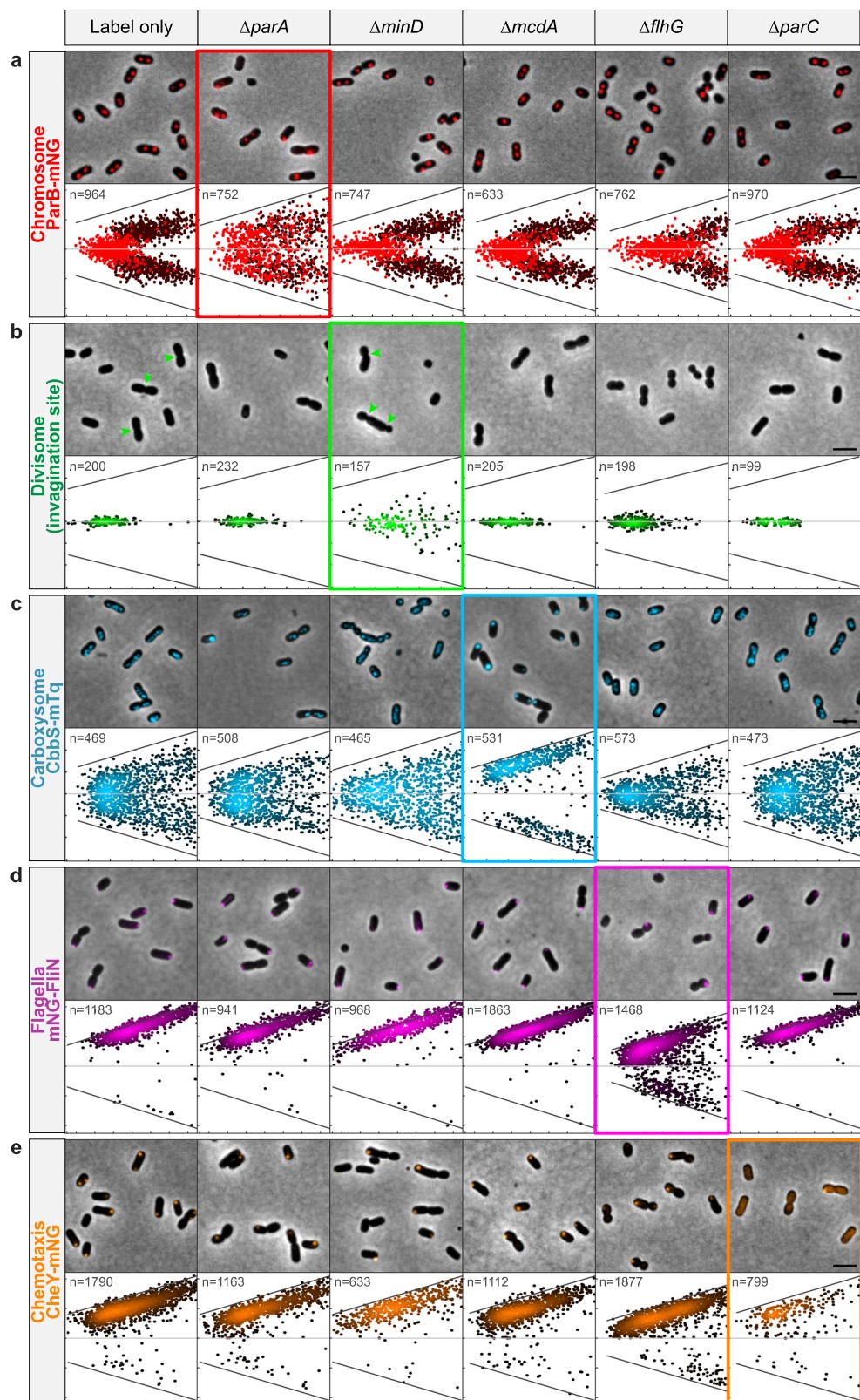

**Fig. 3 | Cargo positioning is not directly controlled by A/D ATPases encoded at distant loci.** Each of the five cellular cargos were fluorescently tagged as indicated (Left): **a** chromosome (ParB-mNG), **b** divisome (invagination site), **c** carboxysomes (CbbS-mTQ), **d** flagella (mNG-FliN), and **e** chemotaxis clusters (CheY-mNG). (All images) Micrographs shown are representative images from at least 3 experiments. Scale bar: 2 µm. The "Label Only" column shows the WT positioning of each of the fluorescent cargos. Deletion of an A/D ATPase resulted in the mislocalization of only its specific cargo (bolded rectangles). Graphs in the "label only" column and graphs in the bolded rectangles are duplicated from Fig. 2. Graph axes for chromosome, carboxysome, flagella, and chemotaxis-labelled mutants: x-axis range (cell length): 0.8–2.1 µm; y-axis range (distance from mid-cell): −1.1–1.1 µm. All x-axes of Δ*flhG* cells are 0.5–1.8 µm. Graph axes for divisome: x-axis range (cell length): 1.5–3.0 µm; y-axis range (distance from mid-cell): −1.5–1.5 µm. Source data are provided as a Source Data file.

influenced in Δ*flhG* cells, albeit with intermediate phenotypes when compared to deleting the dedicated A/D ATPase (Fig. 3, bold boxes). Together our data show that each A/D ATPase is dedicated to the positioning of a specific cargo type, and not *directly* involved in the positioning of other cargos. However, our data at the cell-population level also unveiled potential coordination, crosstalk, and/or interdependencies among certain positioning reactions. In the next three sections, we dissect how cargo positioning by one A/D ATPase can be indirectly coordinated with the positioning and inheritance of a cargo positioned by another A/D ATPase.

### Deletion of *parA*, *minD*, or *flhG* results in anucleate cells via three different mechanisms

We have shown how the deletion of *parA* results in a significant fraction of anucleate cells because ParA is directly required for chromosome segregation and inheritance following cell division (Fig. 4a). Deleting *minD* or *flhG* also resulted in anucleate cells, but the mechanism was different in each case. In Δ*minD* cells, chromosome positioning (Fig. 4b) and ParB foci intensities (Fig. 4c) were similar to that of WT, showing that chromosome segregation was still active. Instead, it was divisome mispositioning in Δ*minD* cells and subsequent asymmetric cell division (Fig. 4d) that indirectly caused asymmetric chromosome inheritance and anucleate cell formation.

In Δ*flhG* cells, chromosome positioning (Fig. 4b) and ParB foci intensities (Fig. 4c) were also similar to that of WT, again suggesting that chromosome segregation was unaffected. But, divisome mispositioning was not as severe as Δ*minD* cells (Fig. 4d, e), suggesting that anucleate cells were forming via a third distinct mechanism. Intriguingly, Δ*flhG* cells were less likely to have two ParB foci compared to WT cells (Fig. 4a). Instead, pre-divisional cells still had a single ParB focus (Fig. 4b) with intensities that suggested the presence of only a single chromosome copy (Fig. 4c). Time-lapse microscopy confirmed that ParB foci positioning was actively maintained (Supplementary Movie 4). Therefore, anucleate Δ*flhG* cells are likely formed due to defects in chromosome replication and/or premature cell division; a question of future study.

Together, the findings emphasize the importance of probing the functional relationships of A/D ATPases with each other and the bacterial cell cycle, when occupying the same organism.

### Anucleate cells inherit carboxysomes

McdA uses the nucleoid as a positioning matrix for distributing carboxysomes[27,28]. Therefore, it was surprising that in the Δ*parA* mutant population, all anucleate cells retained carboxysomes (Fig. 5a). To determine if anucleate cells synthesized carboxysomes de novo or carboxysomes were somehow still inherited following division, we performed time-lapse microscopy on Δ*parA* cells with fluorescent carboxysomes (Fig. 5b and Supplementary Movie 5A). Intriguingly, anucleate cells indeed inherited carboxysomes, but in the most unexpected fashion. In dividing Δ*parA* cells that failed to split their ParB foci, carboxysomes in the to-be-anucleate cell bundled up immediately adjacent to the division plane. Carboxysome bundling was coincident with the chromosome spooling action that occurred at the invaginating septum just prior to complete division and asymmetric chromosome inheritance (see Supplementary Fig. 2g and Supplementary Movie 1B). After septation was complete, the massive carboxysome bundle was explosively liberated from the new pole of the anucleate cell, resulting in multiple freely diffusing carboxysome foci (Fig. 5b and Supplementary Movie 5A). Anucleate cells harboring carboxysomes did not divide further. We found that anucleate cells in the Δ*minD* (Fig. 5c and Supplementary Movie 5B) and Δ*flhG* (Fig. 5d and Supplementary Movie 5C) cell populations also inherited carboxysomes via the same mechanism. Together the data show how carboxysome trafficking and distribution by McdA are dependent upon faithful chromosome segregation. However, anucleate cells can still inherit carboxysomes in *parA*, *minD*, or *flhG* deletion strains because carboxysomes are scraped off of missegregated chromosomes that are spooled through the invagination site during septation. We speculate that all mesoscale cargos using asymmetrically inherited nucleoids as a positioning matrix would show the same mode of inheritance.

### Deletion of *parA*, *minD* or *flhG* influence chemotaxis cluster assembly

The A/D ATPase that positions chemotaxis clusters in *Rhodobacter sphaeroides* uses the nucleoid as its positioning matrix[41]. Therefore, we suspected that A/D ATPase deletions resulting in anucleate cells would indirectly impact the spatial regulation of chemotaxis clusters in these strains. Indeed, we found that *parA*, *minD*, and *flhG* deletion strains, all of which form anucleate cells (see Fig. 4a), had a corresponding increase in cells lacking chemotaxis clusters (Fig. 6a), and when cells had foci, they were notably dimmer compared to WT (Fig. 6b).

It is important to note that while Δ*parA* and Δ*minD* strains exhibited moderate effects on chemotaxis cluster assembly, deletion of *flhG* was more severe (Fig. 6a, b). Given that chemotaxis clusters communicate with the flagellum to move the bacterium towards favorable conditions, we hypothesize that chemotaxis cluster assembly and organization in *H. neapolitanus* is regulated by flagellum positioning. Interestingly, this effect was not reciprocal, as deletion of *parC* had minimal effect on flagella foci number (Fig. 6c) or flagella basal body intensity (Fig. 6d). Identifying the molecular players responsible for this crosstalk in the spatial regulation of the flagellum and chemotaxis cluster is a subject of future work. Together, our data demonstrate interdependencies in how A/D ATPases coordinate the positioning of protein-based organelles with each other as well as the processes of DNA segregation and cell division.

### A/D ATPases have unique interfaces that confer cargo-positioning specificity

We have experimentally demonstrated that multiple A/D ATPases coexist and coordinate the positioning of multiple disparate cargos in the same cell. We also showed that A/D-based positioning is cargo specific. A/D ATPases have been shown to form very similar sandwich dimer structures[45–48], and AlphaFold2 (AF2) predictions suggest this is also the case for all five A/D ATPases of *H. neapolitanus* studied here (Supplementary Fig. 7a-b). The structural similarities suggest there are conserved interfaces unique to each A/D ATPase that provide specificity, linking an A/D ATPase to its particular cargo.

Using AF2 and Rosetta predictions, we determined the A/D ATPase structures from *H. neapolitanus* and identified the putative interaction interfaces that provide specificity to each positioning reaction. There are three interfaces on an A/D ATPase that confer specificity: (1) the dimerization interface, (2) the interface for interaction with its positioning matrix (nucleoid or membrane), and (3) the interface for interaction with its partner protein, which ultimately links the ATPase to its cargo. We identified these three interfaces for all five of the A/D ATPases in *H. neapolitanus* (Fig. 7) and predict key residues required for these associations (Supplementary Data 3).

Specificity at the dimer interface restricts A/D ATPases to homodimerization, and thereby allows each A/D ATPase to function independently in the same cell without cross-interference (Fig. 7a). The putative residues required for homodimerization of the five A/D ATPases in *H. neapolitanus* are provided in Supplementary Data 3, Tab. 1.

ParA-like ATPases use the nucleoid and MinD-like ATPases use the inner membrane for cargo positioning. ParA, McdA, and ParC have basic residues at their C-termini for non-specific binding to nucleoid DNA, whereas MinD and FlhG have membrane targeting sequences (MTSs) for membrane binding (Fig. 7b). We identified the predicted

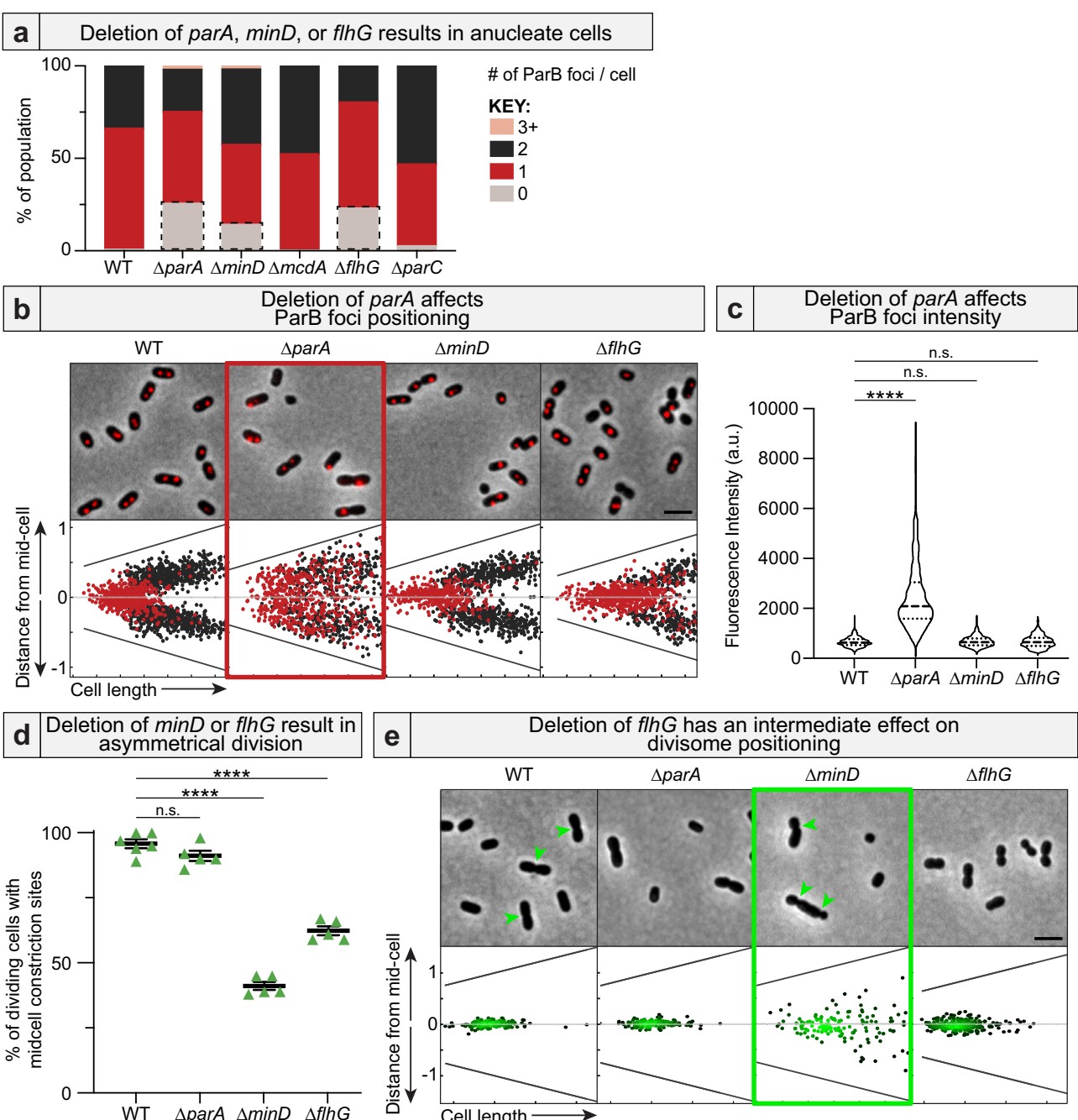

**Fig. 4 | Anucleate cells form via three different mechanisms. a** Deletion of the A/D ATPase required for chromosome (*parA*), divisome (*minD*), or flagellar (*flhG*) positioning resulted in anucleate cells (grey boxes). **b** Only *ΔparA* cells had mispositioned ParB foci. Cells with a single ParB focus are red. Cells with two ParB foci are black. *ΔflhG* cells maintained a single ParB focus in long cells, suggesting a DNA replication defect. **c** ParB foci are brighter only in *ΔparA* cells. WT: $n = 1289$; *ΔparA*: $n = 773$; *ΔminD*: $n = 990$; *ΔflhG*: $n = 747$ biologically independent cells. Kruskal-Wallis p-values: *ΔparA*: <0.0001; *ΔminD*: 0.97; *ΔflhG*: 1. **d** Deletion of *minD* or *flhG* resulted in divisome mispositioning. Constriction sites were considered "mid-cell" when found within 5% of the cell center along the long axis. WT: $n = 6$; *ΔparA*, *ΔminD*, and *ΔflhG*: $n = 5$ biologically independent samples. Brown-Forsythe and Welch ANOVA test p-values: *ΔparA*: 0.28; *ΔminD*: <0.0001; *ΔflhG*: <0.0001 **e** Deletion of *flhG* resulted in an intermediate effect on divisome positioning when compared to *ΔminD*. Each dot on the density graph represents one identified constriction site. Lighter colors represent higher density, darker colors represent lower density. Panels b and e are summarized from Fig. 3. (All images) Scale bar: 2 µm. Source data are provided as a Source Data file.

residues required for each A/D ATPase to associate with their respective positioning matrix (Supplementary Data 3, Tab 2).

Cargo specificity for A/D ATPases comes from interacting with a partner protein that either associates with the cargo, or is a physical component of the cargo itself. Partner proteins have a stretch of amino acids enriched in charged residues at the N-terminus that exclusively interacts with its A/D ATPase, while the rest of the partner protein is

dedicated to cargo association[49]. We generated docking models of each A/D ATPase with an N-terminal peptide from its partner protein (Fig. 7c). The peptides were defined as the first 30 residues of the putative partner protein from the N-terminus. The peptide docking simulations identified several putative residues that are key to system specificity between an A/D ATPase and its partner protein (Supplementary Data 3, Tab 3).

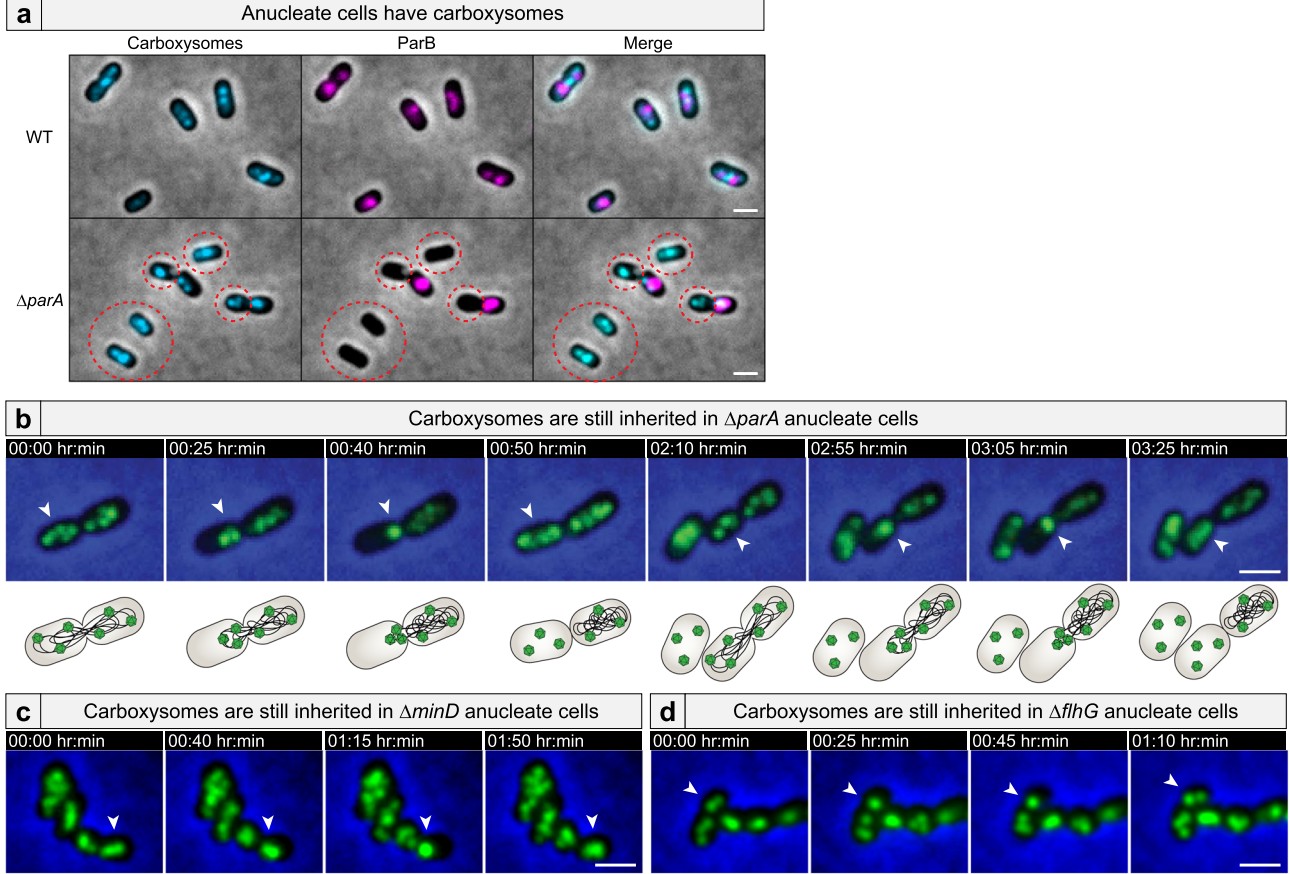

**Fig. 5 | Anucleate cells inherit carboxysomes. a** Carboxysomes are present in anucleate cells (dashed circle). Micrographs shown are representative images from 2 experiments. **b** Time-lapse microscopy shows that carboxysomes (green) in Δ*parA* cells are bound to the nucleoid, but are inherited in anucleate cells via release from the extruded chromosome. Carboxysomes are also inherited in **c** Δ*minD* anucleate cells and **d** Δ*flhG* anucleate cells via the same mechanism. (All videos) White arrows highlight carboxysome bundling and subsequent release. Scale bar: 1 μm.

Finally, we performed in silico alanine-scanning mutagenesis across all residues comprising the interacting interface of the N-terminal peptides of partner proteins docked onto their cognate A/D ATPase (Supplementary Data 3, Tab 4). The resulting ΔΔG values identify the extent to which each residue contributes to the stability of the partner protein interaction with its cognate A/D ATPase. Importantly, our in silico alanine-substitution simulations identified all residues experimentally shown to be important for docking the *Escherichia coli* MinE peptide onto the MinD dimer[47,50,51] (Supplementary Fig. 7c). Together, our in silico data provide a roadmap for strategic mutagenesis and mechanistic probing of the specificity determinants involved in bacterial chromosome segregation, cell division positioning, and protein-based organelle trafficking by A/D ATPases across prokaryotes.

## Discussion

The study of A/D ATPases has focused on a specific cargo of a certain biological process, and largely in model organisms that encode only one or two A/D ATPases. In these studies, two questions are typically posed: How does a specific cargo find its correct position, and how does this position change over time? Here, our focus was on the positioning systems, rather than a specific cargo-type or biological process. Our systems biology approach therefore addresses how multiple A/D ATPases coordinate the positioning of diverse cargos in the same cell.

Encoding multiple A/D ATPases is a shared feature across prokaryotes. We find that over a third of all sequenced bacterial genomes encode multiple A/D ATPases (Fig. 1a), with some bacteria encoding >10. Interestingly, although most bacteria have the same fundamental cargos, not all use dedicated A/D-based positioning systems. For example, many of the cellular cargos we found here to be positioned by A/D ATPases in certain bacteria, like *H. neapolitanus*, are not actively positioned by A/D ATPases in others, like *E. coli*. What necessitates an A/D ATPase for positioning a certain cellular cargo in one bacterium and not in another remains an open question. There does, however, seem to be a limit to the number of A/D ATPases that a bacterium can encode. A/D ATPases are also encoded in archaeal genomes[52], but little is known about their roles in subcellular organization. A recent study showed that archaeal species across several phyla, Euryarchaeota in particular, encode multiple A/D ATPases[53]. Several of these species contained more than a dozen, including *H. volcanii* with 13 A/D ATPases, four of which are MinD-homologs. Strikingly, all four MinD homologs were not required for cell division positioning, but one (MinD4) stimulated the formation of chemotaxis arrays and the archaella, which is the functional equivalent of the bacterial flagellum. This study stresses the importance of experimentally linking A/D ATPases to their cellular cargos as we have done here.

The ParA/MinD family of ATPases spatiotemporally regulates a growing list of diverse mesoscale complexes critical to fundamental processes in prokaryotes, including cell growth and division, DNA segregation, motility, conjugation, and pathogenesis[20,21]. Therefore, understanding how A/D ATPases coordinate the positioning of cellular cargos at the correct location at the correct time is key to understanding bacterial cell function. Using *H. neapolitanus* as a non-pathogenic model, our findings strongly support the idea that each ATPase is dedicated to the positioning of a specific cellular cargo. Our

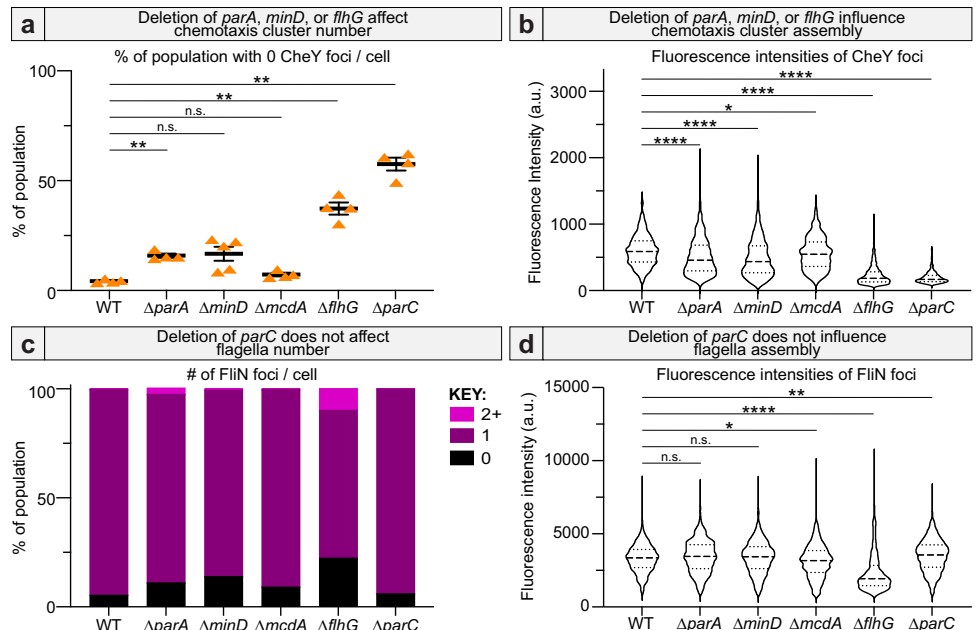

**Fig. 6 | Deletion of *parA*, *minD*, or *flhG* influences chemotaxis cluster assembly.** Deletion of *parA*, *minD*, or *flhG* influences chemotaxis cluster **a** number (WT: *n* = 4, Δ*parA*: *n* = 4, Δ*minD*: *n* = 5, Δ*mcdA*: *n* = 4, Δ*flhG*: *n* = 4, Δ*parC*: *n* = 4 biologically independent samples. Brown-Forsythe and Welch ANOVA test *p*-values: Δ*parA*: 0.0013, Δ*minD*: 0.0642, Δ*mcdA*: 0.1141, Δ*flhG*: 0.0043, Δ*parC*: 0.0012) and **b** assembly (*n* = 455 biologically independent cells. Kruskal-Wallis test *p*-values:

Δ*parA*: <0.0001, Δ*minD*: <0.0001, Δ*mcdA*: 0.0171, Δ*flhG*: <0.0001, Δ*parC*: <0.0001). Only deletion of *flhG* substantially influenced flagella **c** number and **d** assembly (*n* = 851 biologically independent cells. Kruskal–Wallis test *p*-values: Δ*parA*: 0.1161, Δ*minD*: 0.4433, Δ*mcdA*: 0.0184, Δ*flhG*: <0.0001, Δ*parC*: 0.0051). Source data are provided as a Source Data file.

model also unveiled the dependency of protein-based organelle trafficking on the faithful replication and segregation of the chromosome as well as positioning cell division at mid-cell. For example, while the direct consequence of deleting ParA was chromosome missegregation, the resulting asymmetric chromosome inheritance also led to indirect defects in the positioning of carboxysomes and chemotaxis clusters. We also identified a previously uncharacterized epistatic relationship in flagella positioning by FlhG and its downstream influence on the spatial regulation of the chemotaxis cluster by ParC. It is well known that chemotaxis clusters direct cell motility by controlling the direction of flagellar rotation. However, to our knowledge, crosstalk in the spatial regulation of flagella and chemotaxis clusters has yet to be documented. Future studies will determine the molecular players involved in the crosstalk between A/D-based positioning of these two cellular cargos both involved in cell motility.

Our bioinformatics also identified a sixth A/D ATPase in the *H. neapolitanus* genome. *Hn1669* is an A/D ATPase that shows homology to VirC1 and is located near *trb* genes, which encode conjugation machinery components (Fig. 1b). A single study has shown that the VirC1 ATPase, encoded on the Ti plasmid of *Agrobacterium tumefaciens*, is involved in recruiting the conjugative Ti plasmid to the Type IV secretion machine at the cell poles[10]. Intriguingly, in *H. neapolitanus*, the VirC1 homolog is encoded on an Integrative Conjugative Element (ICE) and the downstream gene from this A/D ATPase encodes a ParB homolog. ICEs are major drivers of bacterial evolution and the spread of antibiotic resistance genes[54]. We have yet to image conjugation in *H. neapolitanus*, but we suspect this ParB homolog binds and demarcates the ICE locus during conjugation. It is attractive to speculate that the VirC1-homolog and its downstream ParB homolog are involved in transporting and positioning the ICE locus to the Type IV secretion machine at the cell pole for conjugation.

While A/D ATPases share sequence, structural, and biochemical commonalities, the partner proteins linking these ATPases to their cargos are extremely diverse. Due to this diversity, the partner protein has not always been identified, and as a result, many A/D ATPases

are called 'orphans'[20]. The extreme diversity is largely due to the partner proteins providing the specificity determinants linking an A/D ATPase to its cognate cargo. Despite their extreme diversity, data across the field supports the idea that partner proteins interact with and stimulate their A/D ATPases via a shared mechanism. Partner proteins involved in plasmid partition and chromosome segregation, as well as those required for positioning BMCs, flagella, chemotaxis clusters, and the divisome have all been shown, or suggested, to interact with their A/D ATPase via a positively charged and disordered N-terminus[49]. Our in silico analysis of A/D ATPase dimers docked with N-terminal peptides of their partner proteins demonstrates how specific cargos are assigned, and how these related positioning systems coexist and function in the same cell without cross-interference.

Going forward, we aim to use *H. neapolitanus* as a model to define the general mode of transport shared among the entire A/D ATPase family and to determine how positioning reactions are altered for disparate cargos. These findings are significant because A/D ATPases spatially organize essentially all aspects of bacterial cell function. An additional future direction is to experimentally verify the specificity determinants we identified here for each partner protein and cargo, and leverage this knowledge in the rational design of positioning systems in bacteria. These contributions are expected to be significant because minimal self-organizing systems are vital tools for synthetic biology[55]. We aim to design Minimal Autonomous Positioning Systems (MAPS) consisting of a positioning ATPase and their partner-protein N-terminal peptide as a "luggage tag" to be used as spatial regulators for natural- and synthetic-cargos in heterologous bacteria.

## Methods

### tBLASTn analysis

tBLASTn analysis was done using a ParA/MinD consensus sequence (xKGGxxK[T/S]), as a query against RefSeq Representative genomes database with max target sequences as 5000 and E value threshold at

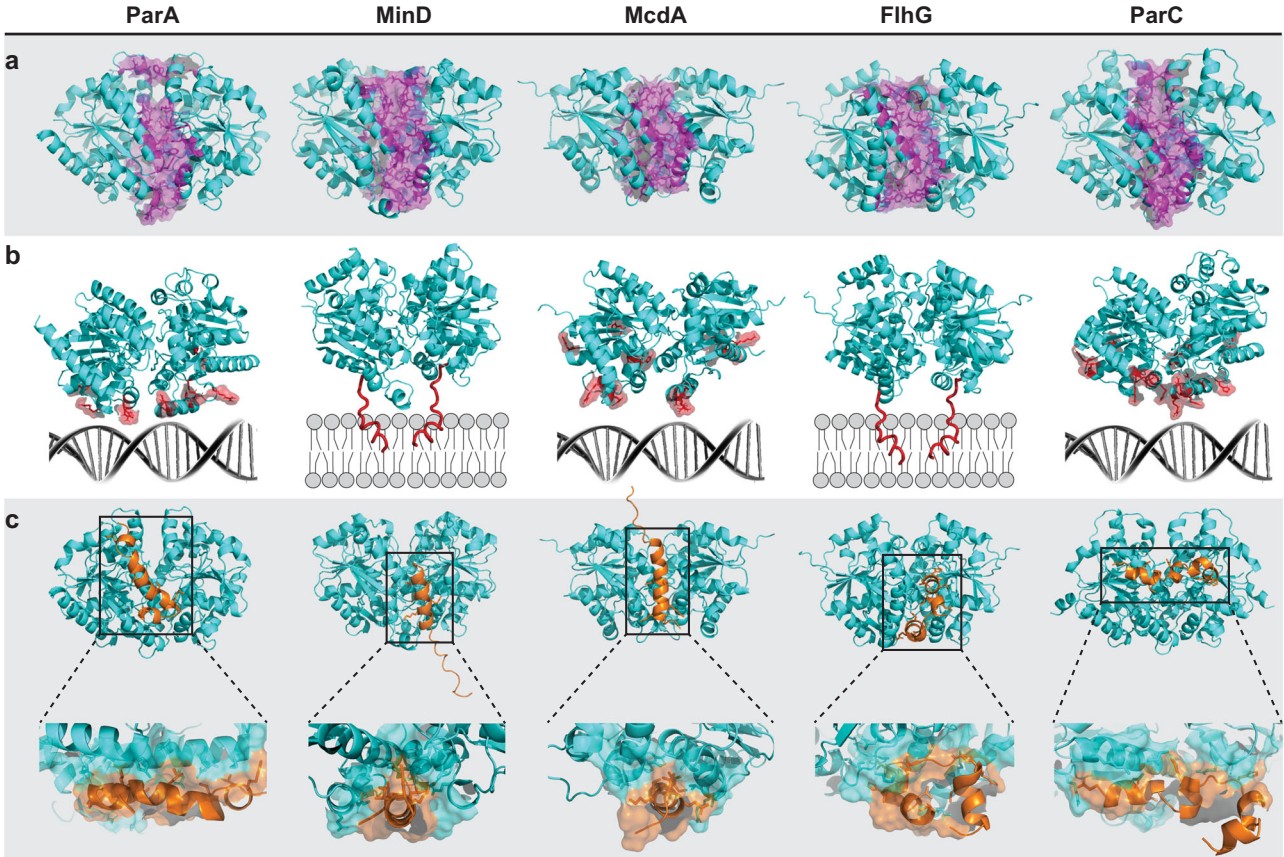

**Fig. 7 | A/D ATPases have unique interfaces that confer cargo-positioning specificity. a** Homodimer structures of the A/D ATPases in *H. neapolitanus* were generated using AlphaFold2 (cyan). Putative residues for homodimer specificity are highlighted magenta. **b** Dimers from a are oriented over their positioning matrix−nucleoid DNA or membrane. Putative residues critical for binding nucleoid DNA or membrane are highlighted red. **c** Dimer structures docked with the N-terminal interacting peptide from putative partner proteins (orange), which confer cargo specificity. Zoomed box: Predicted residues critical for this association are space-filled cyan on the ATPase and orange on the partner protein.

0.0001. Sequences were filtered for those that shared sequence homology and had one of the identified putative cargo genes, confirmed using webFlaGs (https://pubmed.ncbi.nlm.nih.gov/32956448/). The consensus query was generated using COBOLT (https://pubmed.ncbi.nlm.nih.gov/17332019/).

### FlaG analysis
A few representative genomes were selected to display gene neighborhood conservation. Identification of replication origins (OriC's) were performed using Ori-Finder (https://bmcbioinformatics.biomedcentral.com/articles/10.1186/1471-2105-9-79). FlaGs analysis figure was generated using Gene Graphics (https://katlabs.cc/genegraphics/).

### Multiple sequence alignment
Sequences were aligned using Clustal Omega. The resulting tree was imported into iTOL to generate an unrooted tree. (NCBI Accession numbers: Hn2335−ACX97145.1; Hn1364−ACX96198.1; Hn0912−ACX95755.1; Hn0716−ACX95565.1; Hn0722−ACX95571.1; Hn1669−ACX96495.1; Hn0255−ACX95118.1).

### Media and growth conditions
All mutants described in this study were constructed using WT *Halothiobacillus neapolitanus* (Parker) Kelly and Wood (ATCC® 23641™) purchased from ATCC. Cultures were grown in ATCC® Medium 290: S-6 medium for Thiobacilli (Hutchinson et al., 1965) and incubated at 30 °C, while shaken at 130 RPM in air supplemented with 5% CO₂. Strains were preserved frozen at −80 °C in 10% DMSO.

### Construct designs and cloning
All constructs were generated using Gibson Assembly and verified by sequencing. Fragments for assembly were synthesized by PCR or ordered as a gBlock (IDT). Constructs contained flanking DNA that ranged from 750 to 1100 bp in length upstream and downstream of the targeted insertion site to promote homologous recombination into target genomic loci. Cloning of plasmids was performed in chemically competent *E. coli* Top10 or Stellar cells (Takara Bio).

### Making competent cells in H. neapolitanus C2
Competent cells of *H. neapolitanus* were generated as previously reported. In short, 1 L of culture was grown to an OD of 0.1–0.15. Cultures were harvested by centrifugation at 5000 x g for 20 min at 4 °C. Pellets were resuspended and washed twice with 0.5 volumes of ice-cold nanopore water. All wash centrifugation steps were performed at 3000xg for 30 min at 4 °C. The resulting pellet after washing was resuspended in $1 \times 10^{-3}$ volumes of ice-cold nanopore water. These competent cells were used immediately or frozen at −80 °C for future use. Frozen competent cells were thawed at 4 °C before use.

### Transformation in H. neapolitanus C2
50-100 µL of competent cells were mixed with 5 µL plasmid DNA (1–5 µg) and incubated on ice for 5 min. This mixture was then transferred to a tube containing 5 mL ice-cold S6 medium without antibiotics and incubated on ice for 5 min. Transformations were recovered for 16−36 h, while shaken at 130 RPM, at 30 °C, in air supplemented with 5% CO₂. Clones were selected by plating on selective

medium with antibiotics. Colonies were restreaked. Restreaked colonies were verified for mutation by PCR.

## Native fluorescent fusions

For the native fluorescent fusion of ParB-mNG, mNG-FliN, and CheY-mNG, the sequence encoding the fluorescent protein mNeonGreen (mNG) was attached to the 3' or 5' region of the native coding sequences, separated by a GSGSGS linker. For the native fluorescent fusion of Cbbs-mTQ, the sequence encoding the fluorescent protein mTurquoise (mTQ) was attached to the 3' region of the native coding sequence, separated by a GSGSGS linker. A kanamycin resistance cassette was inserted before the gene for N-terminal tags or after the gene for C-terminal tags. When necessary, the promoter was duplicated. The mutant was selected by plating on S6 agar plates supplemented with 50 μg/mL of kanamycin. All fusions were verified by PCR.

## Deletion mutants

For deletions of *Hn2335*, *Hn1364*, *Hn0912*, and *Hn0722*, the genes were replaced with a spectinomycin resistance cassette, followed by a duplicated promoter for the downstream gene. Deletion of *Hn0716* was obtained by codon-optimizing the downstream gene and inserting the spectinomycin resistance cassette after this codon-optimized gene. Mutants were selected by plating on S6 agar plates supplemented with 50 μg/mL of spectinomycin. All mutations were verified by PCR.

## Complementation mutants

The deleted genes (*Hn2335*, *Hn1364*, *Hn0912*, *Hn0716* and *Hn0722*) were individually placed under the expression of a $P_{trc}$ promoter, and inserted into a neutral site, located between genes *Hn0933* and *Hn0934*. Mutants were selected by plating on S6 agar plates supplemented with 25 μg/mL chloramphenicol. The insertion was verified by PCR. For imaging, cells were grown to an OD of 0.1, induced with 0, 0.25, 1, 5, 10, and/or 50 μM IPTG for up to 6 h, and imaged for complementation at various time points. *Hn2335* complemented with 50 μM IPTG for 3 h. *Hn1364* complemented with 50 μM IPTG for 6 h. *Hn0716* complemented with leaky expression of the $P_{trc}$ promoter and induction was not necessary. *Hn0912* and *Hn0722* did not complement with those genes alone. For these two mutants, additional constructs were made to include the neighboring overlapping genes: *Hn0911-Hn0912* and *Hn0722-Hn0723*. Constructs were generated, verified, induced, and imaged as detailed above. *Hn0911-Hn0912* complemented with 50 μM IPTG for 3 h. *Hn0722-Hn0723* complemented with 50 μM IPTG for 3 h.

## Microscopy

All live-cell microscopy was performed using exponentially growing cells. 3-5 μL of cells were dropped onto a piece of 2% UltraPure agarose (Invitrogen, catalog number 16500) + S6 pad and imaged on a 35-mm glass-bottom dish (MatTek, catalog number P35G-1.5-14-C). All fluorescence and phase contrast imaging were performed using a Nikon Ti2-E motorized inverted microscope controlled by NIS Elements software with a SOLA 365 LED light source, a 100X Objective lens (Oil CFI Plan Apochromat DM Lambda Series for Phase Contrast), and a Photometrics Prime 95B Back-illuminated sCMOS camera or a Hamamatsu Orca Flash 4.0 LT + sCMOS camera. ParB-mNG, mNG-FliN, and CheY-mNG were imaged using a "GFP" filter set (C-FL GFP, Hard Coat, High Signal-to-Noise, Zero Shift, Excitation: 470/40 nm [450-490 nm], Emission: 525/50 nm [500-550 nm], Dichroic Mirror: 495 nm). CbbS-mTQ labeled carboxysomes were imaged using a "CFP" filter set (C-FL CFP, Hard Coat, High Signal-to-Noise, Zero Shift, Excitation: 436/20 nm [426-446 nm], Emission: 480/40 nm [460-500 nm], Dichroic Mirror: 455 nm). DAPI fluorescence was imaged

using a standard "DAPI" filter set (C-FL DAPI, Hard Coat, High Signal-to-Noise, Zero Shift, Excitation: 350/50 nm [325-375 nm], Emission: 460/50 nm [435-485 nm], Dichroic Mirror: 400 nm). Alexa Fluor 594 $C_5$ maleimide-conjugated flagella were imaged using a "TexasRed" filter set (C-FL Texas Red, Hard Coat, High Signal-to-Noise, Zero Shift, Excitation: 560/40 nm [540-580 nm], Emission: 630/75 nm [593-668 nm], Dichroic Mirror: 585 nm).

## Long-term microscopy

For multigenerational time-lapse microscopy, 1.5% UltraPure agarose (Invitrogen, catalog number 16500) + S6 pads were cast in 35-mm glass-bottom dishes (MatTek, catalog number P35G-1.5-14-C). Dishes were preincubated at 30 °C in 5% CO2 for at least 24 h. 4 μl of exponentially growing cells were spotted onto the agar pad. Temperature, humidity, and $CO_2$ concentrations were controlled with a Tokai Hit Incubation System. NIS Elements software was used for image acquisition. Cells were preincubated in the stage top for at least 30 min before image acquisition. Videos were taken at one frame per 2.5–5 min for a duration of 12–24 h.

## Image analysis

Several fields of view were captured for each mutant. All fluorescence channels were subjected to background subtraction on Fiji with a rolling ball radius of 50 μm. Background-subtracted fluorescence images were merged with phase contrast images to create composites used for image analysis. Image analysis including cell identification, quantification of cell length, foci localization, foci number, foci fluorescence intensity, and identification of constriction sites were performed using Fiji plugin MicrobeJ 5.13I[56,57]. Cell perimeter detection and segmentation were done using the rod-shaped descriptor with default threshold settings at a tolerance of 56. Maxima detection parameters were individually set for each cargo. For ParB-mNG (cargo = chromosome) foci detection, tolerance and *z*-score were both set to 100. For CheY-mNG (cargo = chemotaxis) foci detection, tolerance was set to 150 and *z*-score was set to 100. For mNG-FliN (cargo = flagella) foci detection, tolerance was set to 710 and *z*-score was set to 69. For CbbS-mTQ (cargo = carboxysome) foci detection, point detection was used instead of foci detection, and tolerance was set to 1010. Results were manually verified using the experiment editor, and non-segmented cells were cut using a particle cutter. Associations, shape descriptors, profiles, and localization were recorded for each strain. Localization graphs were automatically generated through MicrobeJ. Fluorescence intensity graphs and foci number count graphs were made in GraphPad Prism (GraphPad Software, San Diego, CA, www.graphpad.com).

## Nucleoid staining for live imaging

Cells were harvested by centrifugation at 5000 g for 5 min. Following centrifugation, the cells were washed in PBS, pH 7.4. The resulting cell pellet was resuspended in 100 μL PBS and stained with SytoxBlue (Invitrogen, catalog number S34862) at a final concentration of 500 nM. The samples were incubated in the dark, at room temperature for 5–10 min. Stained cells were directly loaded onto an S6 agar pad that had been infused with 500 nM SytoxBlue.

## Carboxysome count

Eleven frames were captured over the course of 2 min. Carboxysomes were counted in each frame. The highest number counted per cell was used for carboxysome count.

## Motility assay

Motility assays were run in S6 in 0.4% agar. Cells were grown on a plate of S6 medium. Individual colonies were inoculated into tubes or plates of motility media and incubated at 30 °C in air supplemented with 5% $CO_2$. Tubes and plates were checked daily for motility for 2 weeks.

## Cysteine-labelling of flagellin

*H. neapolitanus* flagellin was identified using a BLAST homology search with Hag from *Bacillus subtilis* as the query. Only a single flagellin gene was found. Several threonine and serine residues were considered for cysteine mutagenesis. Residues were mutated using the Q5 site-directed mutagenesis kit (NEB, E0554S). Clones were verified for cysteine mutation by sequencing.

## Flagella stain

Alexa Fluor 594 $C_5$ maleimide dye (Invitrogen, A10256) was resuspended in DMSO to the working concentration of 10 mM. *H. neapolitanus* cultures were grown to an OD of 0.1–0.2. The cultures were adjusted to a pH of 7.0 using PBS, pH 11.7. Cells at an adjusted pH of 7.0 were then stained with Alexa Fluor 594 $C_5$ maleimide dye at a final dye concentration of 100 μM. Staining cultures were incubated overnight at 4 °C. The stained cells were washed 4+ times in PBS, pH 7.4. All centrifugation steps were performed at 5000 × g for 3 min.

## Statistical analysis

For whole population analyses, such as fluorescence intensity and cell length measurements, we performed a non-parametric Wilcoxon (Kruskal-Wallis) test followed by a Dunn's multiple comparisons test. For percent of population analyses, such as foci number or the presence of mid-cell constriction sites, populations were analyzed as separate field of views and the summary values were plotted as separate data points. We then performed a Brown-Forsythe and Welch ANOVA test followed by Dunnett's T3 multiple comparisons test. GraphPad Prism (version 9.5.1) was used to perform all analyses. *P*-value style: 0.1234 (ns), 0.0332 (*), 0.0021 (**), 0.0002 (***), < 0.0001 (****).

## Movie editing

Movies were cropped using Fiji. Movies were stabilized using the multi-channel hyperstack alignment plug-in called HyperStackReg. Time stamp and scale bar annotations were added using Fiji. Arrows and pauses were added using Adobe Premier Pro.

## Structure prediction and peptide docking using AlphaFold2 and Rosetta

For MinD and McdA, we generated the N-terminus peptide-ATPase docking models using the CollabFold implementation of AlphaFold2[58,59]. The N-terminal peptides were defined as the first 30 residues of the putative partner protein from the N-terminus. Multiple sequence alignments were constructed using MMseqs2. For each peptide-ATPase pair, we generated five structures with the default CollabFold/AlphaFold2 hyperparameters saved for the number of recycles for each model being increased to 12. As default to Alphafold2, the structures were energetically minimized with AMBER using the Amber99sb force field. We selected docked peptide models based on the pLDDT scores of the binding interface residues and similarity to previously resolved ParA-like ATPase/partner-protein crystal structures.

For ParA, FlhG, and ParC we generated docked peptide models using Rosetta's FlexPepDock protocol[60]. First, we equilibrated the ATPase homodimer structures generated from AlphaFold2 to the ref2015_cart_cst Rosetta force field with the FastRelax full-atom refinement protocol with cartesian coordinate space minimization using the lbfgs_armijo_nonmonotone minimizer. To preserve the position of the backbone atoms predicted by AlphaFold2, a backbone atom coordinate constraint was added. For this initial step, we generated 20 trajectories with the lowest scoring structure being used for the peptide docking step. We used the score3 with the docking_cen.wts_patch and the REF2015 force fields for the low-resolution and high-resolution docking steps of the FlexPepDock protocol, respectively. For each putative partner protein/ATPase pair, we simulated 50,000 docking trajectories. The top two thousand trajectories defined by the lowest energetics were then clustered using Calibur[61].

We then chose the final models based on cluster information, energetics, and mechanistic plausibility. From the final docked models chosen from the Rosetta and Alphafold2 simulations, we sought to identify key binding residues through an *in-silico* alanine mutation scan and ΔΔG calculations. We iteratively mutated all interface residues of the docked peptide and calculated the ΔΔG using the FlexDDG protocol in Rosetta with the talaris2014 forcefield[62]. In FlexDDG, for each mutation, the backbone and side chain conformations were sampled 35,000 times using Rosetta's Monte Carlo backrub method. At every 2500 sample interval the ΔΔG of mutation was calculated. The final reported ΔΔG is the average of 35 such trajectories.

(NCBI Accession numbers: ParA partner protein (ParB)− ACX97144.1; MinD partner protein (MinE)−ACX96199.1; McdA partner protein (McdB)−ACX95754.1; FlhG partner protein (FliA)−ACX95566.1; ParC partner protein (CheW)−ACX95572.1). The first 30 amino acids from the N-terminus of each partner protein were docked onto the dimer structures in silico using AlphaFold2 (MinD, McdA) or Rosetta (ParA, FlhG, ParC). Binding interface was defined by those residues that shared a minimum of 1 Angstrom² of surface area.

## Reporting summary

Further information on research design is available in the Nature Portfolio Reporting Summary linked to this article.

## Data availability

Bacterial genomes obtained from the NCBI RefSeq DataBase (https://www.ncbi.nlm.nih.gov/refseq/). Experimentally determined ParA/MinD ATPase structures obtained from the Protein Data Bank (PDB): ParA (5U1G); McdA (6NOP); ParC (5U1G); MinD (3Q9L); FlhG (4RZ3). All data generated or analyzed during this study are included in this published article and its supplementary information files. Source data are also provided with this paper. Source data are provided with this paper. All data generated or analyzed during this study are included in this published article and its supplementary information files. Source data are also provided with this paper. Source data are provided with this paper.

## Code availability

The code generated during this study are available at GitHub: Identifying ParA/MinD ATPases in bacteria using BLAST (https://github.com/krthkkrv/Multiple-ParA-MinD-ATPases-coordinate-the-positioning-of-disparate-cargos-in-a-bacterial-cell)[63], Identifying specificity determinants of each ParA/MinD ATPase in *H. neapolitanus* using AlphaFold2 and Rosetta (https://github.com/jilimcaoco/Multiple-ParA-MinD-ATPases)[64].

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

## Acknowledgements

We thank Joshua S. MacCready, Christopher A. Azaldegui, Joseph L. Basalla, Ajai J. Pulianmackal, and Claudia Mak for their thoughtful discussions. We thank Holly Turula and Miles Mckenna for their technical assistance. This work was supported by the National Science Foundation Award No. 1817478 (A.G.V.), the National Science Foundation Graduate Research Fellowship Program DGE 1841052 (L.T.P.), and from research initiation funds provided by the MCDB Department (A.G.V.).

## Author contributions

Conceptualization, L.T.P. and A.G.V.; Methodology, L.T.P., M.J.L., K.R.; Formal Analysis, L.T.P. M.J.L., K.R., S.Y., M.G., M.J.O. and A.G.V.; Investigation, L.T.P., J.Z., and M.K.T.; Resources, A.G.V.; Writing—Original Draft, L.T.P. and A.G.V.; Visualization, L.T.P., J.M.I.L., K.R., and A.G.V.; Supervision, L.T.P., M.J.O., and A.G.V.; Funding Acquisition, L.T.P. and A.G.V.

## Competing interests

The authors declare no competing interests.
