## [Peer Review File · Nature Communications]

Multiple ParA/MinD ATPases coordinate the positioning of disparate cargos in a bacterial cellReviewer #1 (Remarks to the Author):

The manuscript by Pulianmackal and colleagues, entitled "Multiple ParA/MinD ATPases coordinate the positioning of disparate cargos in a bacterial cell", provide a comprehensive and in-depth examination of five members of the ParA/MinD (A/D) family of ATPases in a slow growing bacterium, *Halothiobacillus neapolitanus*. The authors employ a multi-disciplinary approach, incorporating bioinformatics, deletion analyses, in vivo functional imaging and in silico techniques to investigate the specific functions of each ATPase and their interactions with various cargos. The bioinformatics survey of the A/D ATPase family in the NCBI database provides a valuable foundation for the study showing that ~95% of the bacteria present in the NCBI database encode at least one A/D ATPases and ~30% multiple ones. The focus on the bacterium encoding seven A/D ATPases further emphasizes the significance of this family of ATPases in bacterial physiology. The authors then focus their study on a bacterium that encode 7 A/D ATPases and carefully demonstrate that 5 of them are indeed specifically implicated in the localization of 5 specific cargos, namely chromosome (OriC region), divisome, carboxysomes, flagella and chemotaxis clusters. A systematic analysis of the positioning and dynamics of all cargos in all single deletions background shows that the positioning of OriC, the divisome and the chemotaxis cluster are indirectly affected by the deletion of the distant FlhG ATPase. They characterize three different mechanisms resulting in anucleate cells mediated by the deletions of parA, minD or flgH, and then solve the surprising finding that all anucleate cells harbor carboxysome. Lastly, they analyse, using prediction tools (and compare with existing structural data for some close homologs), the three interfaces of the 5 A/D ATPases for self-dimerisation, interaction with the matrix (DNA or membrane) and interaction with the cargo. This latter is investigate by docking with the first 30 AA of the proposed partner, and these interactions are tested by in silico alanine-scanning mutagenesis.

The manuscript is well written, and the experiments are well-designed and performed with appropriate controls. The results support the conclusions, and the figures are well-organized and annotated. The authors provide clear explanations of their findings and provide a logical and coherent storyline.

I have only very few minor suggestions.

L. 149: "when present" must read "when absent"?

L. 258: it is not obvious that the positioning of the chemotaxis clusters are influenced in a delta flhG" background. Please provide more explanation and possibly quantification.

Figure 2: the legend for all graphs should be inverted between x- and y- axes. Or may be the orientation should be kept as for fig. 3 & 4.

Also, color code for dots (colors versus black) should be indicated.

Reviewer #2 (Remarks to the Author):

Many species of bacteria possess more than one system of ParA/MinD family ATPases, which are involved in cellular organization processes. The functions and mechanisms of these systems have been studied in detail, albeit only at the single-system level and not in the context of the other systems that might be active in the cell. In this manuscript the authors addressed this question. They started out by a global analysis of bacterial genome sequences and found that more than a third of bacteria analyzed harbor multiple ATPases of the ParA/MinD family. To tackle the question of how these different system have interfering functions they chose a suitable model species, which was, however, none of the usual text book models but the sulfur-oxidizing gammaproteobacterium *Halothiobacillus neapolitanus*, which encodes seven putative A/D-ATPases. In silico analysis of the genetic context indicated that these A/D-ATPases are involved in conjugation, nitrogen metabolism, chromosome segregation (ParA), divisome placement (MinD), carboxysome arrangement and segregation (McdA), flagellum formation (FlhG) and localization of the chemotaxis array (ParC). By microscopic and physiological analysis of the corresponding gene deletion mutants, the authors then convincingly demonstrated that the latter five systems in fact have the predicted functions. To enable an identification of a potential functional cross talk

between the A/D-ATPases, the authors characterized the response of one specific system's 'cargo' in the absence of each single other A/D-ATPases. They found that, while each system's function is not directly affected by the other A/D-ATPase systems, there is some indirect cross talk as loss of parA, minD, or flhG affects chemotaxis cluster assembly. The high-quality microscopy approach revealed further interesting observations, e.g., that loss of the three latter proteins leads to formation of anucleate cells, which however still inherit carboxysomes. A corresponding mechanism is proposed. Finally, the authors performed an in silico comparison of the A/D-ATPase proteins structures and identified regions that likely define the specificity for the corresponding 'cargos' and targets. These predictions were not further followed up by experimental approaches.

In this study, the authors appropriately addressed a highly important question. I am deeply impressed by the effort to use non-standard model system, *H. neapolitanus*, which required a true tour-de-force to first show that the addressed A/D-ATPases have, in fact, their designated role. The quality of the microscopy data is great. While the overall outcome of the study may not be surprising, there are a number of novel observations and further question that can be addressed. The manuscript itself was well structured and I enjoyed reading it. There were however, some issues, that the authors should consider.

1. In general: Given the amount of the data presented, I imagine that the authors ran into limitations with respect to the overall length of the manuscript. However, there are still several points that I miss in the discussion section. For example, for all of the A/D-systems the authors have looked into, there are known client proteins, which need to be mentioned. How do the systems work and how does this relate to, e.g., the predicted interaction surfaces with the 'cargo', there is a whole lot that can (and should) be discussed. I also miss the beautiful studies on the role of FlhG taking over for Min in *Campylobacter*. I therefore wonder about the relatively small number of references of original studies. This is also true for the Materials & Methods section, which should get some more references where appropriate.

2. I am not too sure about the analysis of proteins structures and the following predictions. It is great that the authors performed this analysis and put it out. However, without any actual mutant testing it remains a prediction, which would turn into a beautiful study on its own with some more experiments.

3. As far as I can see, the deletion mutants have been constructed by integration of a resistance cassette, but I have not seen any complementation controls - I am stressing this because of the following point 4. I am also missing some stability and functional controls for the fluorescently tagged client proteins or other mutants (such as the Cys-bearing flagellin).

4. I have some questions about the role of Hn0716/flhG, also because many of the original findings described in this manuscript relate to an flhG mutant. The authors state that FlhG typically mediates flagellar number and position (280 ff). In polarly flagellated gammaproteos, FlhG (or FleN) almost always regulates flagellar number (so far, so good), but polar positioning of the flagellum rather depends on the GTPase FlhF, which has also been described to be involved in placing the chemotaxis cluster (for example shown for *Pseudomonas aeruginosa*). flhF is always located upstream of and often overlapping with flhG, and some of the phenotypes the authors describe are reported for flhF flhG double mutants. While the authors may be completely right with their findings (making them even more interesting), I strongly urge them to double check the Hn0716/flhG mutant including complementation.

Same chapter: The authors show that in the absence of Hn0716/flhG, clusters of the C-ring protein FliN get misplaced and the fluorescence signal of these foci gets weaker. At the same time, the authors show that frequently flagellar tufts rather than single flagella are formed at these positions. Typically, the formation of more flagellar systems means more basal bodies including the C-ring proteins. More flagellar filaments with less C-ring signal does not really fit, hence the request for a stability check of the fluorescent fusion), unless FlhG also has a role in C-ring assembly. There is some work on this, which should be considered and discussed.

5. I have also some questions about Hn0722/parC. In polarly flagellated gammaproteobacteria, the chemotaxis array is almost always placed close to the polar flagellum. A role of Par proteins in assembly and positioning has been addressed for *Vibrio*, where positioning depends on a polar

landmark protein. In *P. aeruginosa*, the GTPase FlhF may be involved (as mentioned above). Chemotaxis clusters may also be loaded on and distributed to the daughter cells by the chromosome (e.g. in *Rhodobacter*), which appears to be the model the authors prefer. However, from the experiments this is not clear and the studies on the different systems need to be taken into account and be discussed.

6. 256, what about carboxysome amounts in MinD mutants? Please elaborate.

Other points:

3, just out of interest, I thought that the type IV pili machinery is considered a linear motor, isn't that true?

102, the authors mention that also some pathogens have a set-up with several A/D-ATPases? If this is of particular interest (why should it), these species should be named.

120, is FlaG a commonly used abbreviation? I instantly think of flagellar genes (but that maybe only me).

238, do the authors really mean swarming across a surface? Or spreading in soft agar?

318, 6B, significance for ParA and MinD?

Reviewer #3 (Remarks to the Author):

This is a paper full of useful information on the ParA/MinD family of proteins and will serve as an important resource for the scientific community. The work is extensive, well done, well presented and rich in information. The authors do an extensive bioinformatic analysis of this protein family and document the existence of members of this family throughout the bacterial world. I commend the authors for putting together such a complete picture and not trying to publish this piecemeal.

One comment, however, is that the authors set up somewhat of a false premise by wondering about whether "multiple A/D ATPases can coexist and coordinate the positioning of such diverse cargos in the same cell." (lines 10-12 in the abstract). The predictable answer to what is required is summarized nicely between lines 341-4: 1) dimerization interface of the ATPase; 2) interaction with the matrix; and 3) link between ATPase and cargo. I say predictable because of the work already done with MinD, ParA and the authors work on McdA. It seems like the only way these might interfere with each other is if they competed for the same matrix, which is unlikely due to their small number of molecules per cell. Having said that this work establishes the fact that so many family members can readily coexist in the same cell and the bioinformatics and in silico predictions reinforces the principles laid out between lines 341-4 as mentioned above.

Other comments:

The extensive tables indicate how many A/D proteins are present in each organism but omit those, if any exist (~5%; line 85), that do not have any. Does that mean that there are convincing chromosome sequences available are there any that have none? Is it 5%? Could they be included in the Table? Or another way to ask it is of the organisms in the Table what fraction have MinD or ParA for example. We already know *E. coli* lacks ParA and there clearly are organisms that lack MinD or others. Could this information be added to the Table? So for example, if an organism has 5 unique ATPases what are they?

There are several interesting observations about this organism as the authors point out and are the subject of future studies. ParA affects chromosome segregation resulting in anucleate cells but

has almost no effect on division. This is clearly quite different than what occurs in *Caulobacter*, for example, where I believe ParA is essential. Also, it may be that bacteria that do multifork replication do not have ParA (*Caulobacter*, *Vibrio*) whereas those that do, do not have ParA (*E. coli*, *Salmonella*, *Bacillus* etc.)

Line 82. I would change 'can be' to 'are'

Line 139. Chromosome segregation in most bacteria. Not sure if this is true. *Caulobacter* and *Vibrio* have ParA but *E. coli* and *B. subtilis* do not (actually *B. subtilis* has Soj but is quite different than ParA).

Line 280-285 and Fig. 3. Δ FlhG-clearly has delayed ParB foci (Fig. 4B) but division does not appear delayed.

Point-by-Point Response:

Reviewer #1 (Remarks to the Author):

The manuscript by Pulianmackal and colleagues, entitled "Multiple ParA/MinD ATPases coordinate the positioning of disparate cargos in a bacterial cell", provide a comprehensive and in-depth examination of five members of the ParA/MinD (A/D) family of ATPases in a slow growing bacterium, *Halothiobacillus neapolitanus*. The authors employ a multi-disciplinary approach, incorporating bioinformatics, deletion analyses, in vivo functional imaging and in silico techniques to investigate the specific functions of each ATPase and their interactions with various cargos.

The bioinformatics survey of the A/D ATPase family in the NCBI database provides a valuable foundation for the study showing that ~95% of the bacteria present in the NCBI database encode at least one A/D ATPases and ~30% multiple ones. The focus on the bacterium encoding seven A/D ATPases further emphasizes the significance of this family of ATPases in bacterial physiology.

The authors then focus their study on a bacterium that encode 7 A/D ATPases and carefully demonstrate that 5 of them are indeed specifically implicated in the localization of 5 specific cargos, namely chromosome (OriC region), divisome, carboxysomes, flagella and chemotaxis clusters. A systematic analysis of the positioning and dynamics of all cargos in all single deletions background shows that the positioning of OriC, the divisome and the chemotaxis cluster are indirectly affected by the deletion of the distant FlhG ATPase. They characterize three different mechanisms resulting in anucleate cells mediated by the deletions of parA, minD or flhG, and then solve the surprising finding that all anucleate cells harbor carboxysome. Lastly, they analyse, using prediction tools (and compare with existing structural data for some close homologs), the three interfaces of the 5 A/D ATPases for self-dimerisation, interaction with the matrix (DNA or membrane) and interaction with the cargo. This latter is investigated by docking with the first 30 AA of the proposed partner, and these interactions are tested by in silico alanine-scanning mutagenesis.

The manuscript is well written, and the experiments are well-designed and performed with appropriate controls. The results support the conclusions, and the figures are well-organized and annotated. The authors provide clear explanations of their findings and provide a logical and coherent storyline.

We thank the reviewer for the kind words and shared excitement for this work!

I have only very few minor suggestions.

L. 149: "when present" must read "when absent"?

Clarified as follows (changes in red): "*When the putative parA gene (Hn2335) was deleted, ParB-mNG foci were completely absent in 25% of cells (Supplementary Figure 2d). When ParB foci were present, they were randomly positioned regardless of cell length (Figure 2c) and significantly brighter compared to that of WT (wild-type) cells (Supplementary Figure 2e).*" (Line 138)

L. 258: it is not obvious that the positioning of the chemotaxis clusters are influenced in a delta flhG" background. Please provide more explanation and possibly quantification.

The reviewer is correct. We have clarified the statement as follows:

"Intriguingly, chromosome (Figure 3a), divisome (Figure 3b), and chemotaxis cluster (Figure 3e) positioning or focus intensity were all influenced in Δ flhG cells, albeit with intermediate phenotypes when compared to deleting the dedicated A/D ATPase (Figure 3, bold boxes)." (Line 271)

Figure 2: the legend for all graphs should be inverted between x- and y- axes. Or maybe the orientation should be kept as for fig. 3 & 4.

Thank you for noticing this! The axes have been corrected in the legend.

Also, color code for dots (colors versus black) should be indicated.

We now include the following statement in the legend explaining the density plots: "*For all density plots, lighter colors represent higher density, darker colors represent lower density.*" (Line 950)

Reviewer #2:

Many species of bacteria possess more than one system of ParA/MinD family ATPases, which are involved in cellular organization processes. The functions and mechanisms of these systems have been studied in detail, albeit only at the single-system level and not in the context of the other systems that might be active in the cell. In this manuscript the authors addressed this question. They started out by a global analysis of bacterial genome sequences and found that more than a third of bacteria analyzed harbor multiple ATPases of the ParA/MinD family. To tackle the question of how these different system have interfering functions they chose a suitable model species, which was, however, none of the usual text book models but the sulfur-oxidizing gammaproteobacterium *Halothiobacillus neapolitanus*, which encodes seven putative A/D-ATPases. In silico analysis of the genetic context indicated that these A/D-ATPases are involved in conjugation, nitrogen metabolism, chromosome segregation (ParA), divisome placement (MinD), carboxysome arrangement and segregation (McdA), flagellum formation (FlhG) and localization of the chemotaxis array (ParC). By microscopic and physiological analysis of the corresponding gene deletion mutants, the authors then convincingly demonstrated that the latter five systems in fact have the predicted functions. To enable an identification of a potential functional cross talk between the A/D-ATPases, the authors characterized the response of one specific system's 'cargo' in the absence of each single other A/D-ATPases. They found that, while each system's function is not directly affected by the other A/D-ATPase systems, there is some indirect cross talk as loss of *parA*, *minD*, or *flhG* affects chemotaxis cluster assembly. The high-quality microscopy approach revealed further interesting observations, e.g., that loss of the three latter proteins leads to formation of anucleate cells, which however still inherit carboxysomes. A corresponding mechanism is proposed. Finally, the authors performed an in silico comparison of the A/D-ATPase proteins structures and identified regions that likely define the specificity for the corresponding 'cargos' and targets. These predictions were not further followed up by experimental approaches.

In this study, the authors appropriately addressed a highly important question. I am deeply impressed by the effort to use non-standard model system, *H. neapolitanus*, which required a true tour-de-force to first show that the addressed A/D-ATPases have, in fact, their designated role. The quality of the microscopy data is great. While the overall outcome of the study may not be surprising, there are a number of novel observations and further question that can be addressed. The manuscript itself was well structured and I enjoyed reading it. There were however, some issues, that the authors should consider.

We thank the reviewer for their kind words and critical assessment of our work.

1. In general: Given the amount of the data presented, I imagine that the authors ran into limitations with respect to the overall length of the manuscript. However, there are still several points that I miss in the discussion section. For example, for all of the A/D-systems the authors have looked into, there are known

client proteins, which need to be mentioned. How do the systems work and how does this relate to, e.g., the predicted interaction surfaces with the 'cargo', there is a whole lot that can (and should) be discussed. I also miss the beautiful studies on the role of FlhG taking over for Min in *Campylobacter*. I therefore wonder about the relatively small number of references of original studies. This is also true for the Materials & Methods section, which should get some more references where appropriate.

We thank the reviewer for their shared excitement and interest regarding the mechanisms and client proteins associated with each of the five cargoes studied here. Earlier versions of the manuscript provided a more detailed and nuanced view of our findings that focused on the specifics of each positioning system and associated cargo. However, as the reviewer appreciates, the manuscript was prohibitively long and previous reviewers found these details detracted from the focus of this work – how multiple systems *coordinate* in their positioning reactions. Given the 30+ years of work on Par and Min, it would be difficult to give these two systems (and researchers) due justice because of space limitations. We are working on a detailed review focused on comparing and contrasting the positioning systems from these disparate cargoes. However, we agree that we can do better here for the relatively less studied FlhG ATPase and its role in organizing flagella. We now dedicate a Supplementary Discussion that puts our FlhG results into context with previous work in the field (and associated citations).

2. I am not too sure about the analysis of protein structures and the following predictions. It is great that the authors performed this analysis and put it out. However, without any actual mutant testing it remains a prediction, which would turn into a beautiful study on its own with some more experiments.

We agree with the reviewer that the *in silico* work is solely predictive at this point, and we label them as such in the paper. The manuscript currently highlights published data from the *E. coli* Min system to strongly support the predictive power of this technique. We also state these predictions serve as a useful road map for strategic mutagenesis experiments for the field, and ourselves. Indeed, we have moved forward to experimentally verify the importance of N-terminal residues predicted to be important for McdB associations with McdA. Our preliminary data indeed further verifies the *in silico* predictions, but not in a straightforward manner - some of the predicted mutations destroy the association, some have intermediate phenotypes, some are neutral on their own but become important if other residues are also mutated, and some actually strengthen the association with McdA. We believe these studies have not only identified key residues on McdB required for interaction with McdA, but also distinguished the catalytic residues that stimulate McdA ATPase activity! We believe these data are better suited towards an upcoming publication that focuses specifically on the mechanism of McdA/McdB association. As the reviewer notes, we hope that our lab (and others) will use these predictions for studies focused on specific positioning reactions associated with each cargo.

3. As far as I can see, the deletion mutants have been constructed by integration of a resistance cassette, but I have not seen any complementation controls - I am stressing this because of the following point 4. I am also missing some stability and functional controls for the fluorescently tagged client proteins or other mutants (such as the Cys-bearing flagellin).

Complementation controls are now provided for all A/D ATPase deletion mutants in their respective supplementary figures (S2H, S3J, S4H, S5I, S6F) and associated description in the main text (Lines 142, 176, 194, 220, 255).

We understand the reviewer's request of further experiments that probe the function and stability of the fluorescently-tagged client proteins and the Cys-bearing flagellin mutant. However, we do not think these controls are essential here because we are performing relative comparisons in the presence and absence of the positioning ATPase. In other words, even if there are subtle differences in stability and function of the fluorescent fusions, the differences in the presence and absence of the ATPase remain meaningful.

Going forward, we believe such experiments will be very important when probing specific positioning mechanisms associated with each cargo.

4. I have some questions about the role of Hn0716/flhG, also because many of the original findings described in this manuscript relate to an flhG mutant. The authors state that FlhG typically mediates flagellar number and position (280 ff). In polarly flagellated gammaproteos, FlhG (or FlhN) almost always regulates flagellar number (so far, so good), but polar positioning of the flagellum rather depends on the GTPase FlhF, which has also been described to be involved in placing the chemotaxis cluster (for example shown for *Pseudomonas aeruginosa*). flhF is always located upstream of and often overlapping with flhG, and some of the phenotypes the authors describe are reported for flhF flhG double mutants. While the authors may be completely right with their findings (making them even more interesting), I strongly urge them to double check the Hn0716/flhG mutant including complementation.

We thank the reviewer for bringing up these very interesting details that relate to FlhG function in *H. neapolitanus* and how the phenotype of not just hyperflagellation, but also mispositioning, is different from previously studied gammaproteobacteria. In *H. neapolitanus*, a putative *flhF* gene is indeed upstream, but not overlapping the *flhG* gene (See Figure S1, Panel D). In our newly added complementation assays (suggested by this reviewer, and thank you), we re-introduced just the *flhG* gene (expressed from an exogenous locus) and obtained complete recovery of flagella number positioning. The additional data strongly supports that, unlike other gammaproteobacteria studied thus far, deletion of *H. neapolitanus flhG* alone results in both mispositioning and hyperflagellation. We agree with the reviewer that this difference makes the finding even more interesting. We now include text highlighting this difference in the results, and in a dedicated a “Supplementary Discussion” that puts our FlhG results into context with previous work in the field (and associated citations).

Same chapter: The authors show that in the absence of Hn0716/flhG, clusters of the C-ring protein FlhN get misplaced and the fluorescence signal of these foci gets weaker. At the same time, the authors show that frequently flagellar tufts rather than single flagella are formed at these positions. Typically, the formation of more flagellar systems means more basal bodies including the C-ring proteins. More flagellar filaments with less C-ring signal does not really fit, hence the request for a stability check of the fluorescent fusion), unless FlhG also has a role in C-ring assembly. There is some work on this, which should be considered and discussed.

Please see our comments to points 3 and 4. We agree this is an interesting new finding that requires further experimentation. Since these questions fall specifically on understanding flagellar spatial organization, we aim to address these and other related questions in a future study. However, we now bring up these interesting differences in a “Supplementary Discussion”. We thank the reviewer for their critical reading and expert analysis of FlhG function. We believe these additions have made the paper considerably stronger.

5. I have also some questions about Hn0722/parC. In polarly flagellated gammaproteobacteria, the chemotaxis array is almost always placed close to the polar flagellum. A role of Par proteins in assembly and positioning has been addressed for *Vibrio*, where positioning depends on a polar landmark protein. In *P. aeruginosa*, the GTPase FlhF may be involved (as mentioned above). Chemotaxis clusters may also be loaded on and distributed to the daughter cells by the chromosome (e.g. in *Rhodobacter*), which appears to be the model the authors prefer. However, from the experiments, this is not clear and the studies on the different systems need to be taken into account and be discussed.

We thank the reviewer for bringing up this important point. *H. neapolitanus* does not encode HubP, TipN, or any other obvious polar landmark protein. It remains possible that there is a novel polar landmark protein that fulfills the role of HubP or TipN in *H. neapolitanus*. However, it is important to note that the

chemotaxis arrays in *H. neapolitanus* are not at the extreme cell pole as they are in *Vibrio*. Instead, they are polar-adjacent in *H. neapolitanus*. We believe this is also an important distinction that warrants further research. We have made improvements to better highlight how ParC/PpfA-like ATPases work in other organisms and added the associated references: “*In Vibrio species, ParC directs chemotaxis arrays to a polar landmark protein called HupB*^{8,41}. *As a consequence, daughter cells inherit an array at their old pole upon cell division. In R. sphaeroides, there are no polar landmarks and PpfA distributes chemotaxis clusters over the nucleoid*^{9,40}. *Where studied, deletion of the A/D ATPase alters chemotaxis cluster number and positioning in cells, which results in a reduction in spreading motility. Our bioinformatics analysis showed that the protein encoded by Hn0722 is a ParC/PpfA homolog within the chemotaxis operon of H. neapolitanus (see Figure 1C).*” (Lines 242-245)

Future studies we will specifically focus individual positioning systems and their respective cargo.

6. 256, what about carboxysome amounts in MinD mutants? Please elaborate.

Thank you for pointing out the differences in cell numbers. The carboxysome numbers looked fewer in MinD mutants because there were fewer cells in the FOVs. However, the sample size for the MinD strain is still sufficient to achieve a confidence level of 95% that the real value is within $\pm 5\%$ of the measured values. But to prevent readers from over interpreting this difference we have now updated the graphs to have similar numbers of cells for across all strains.

Other points:

3, just out of interest, I thought that the type IV pili machinery is considered a linear motor, isn't that true?

Good catch! Thank you. We have made the following minor adjustments to the text: “*In bacteria, where linear motors involved in spatial regulation are absent, the ParA/MinD (A/D) family of ATPases organize an array of genetic- and protein-based cellular cargos.*” (Line 3) And “*In bacteria, however, where linear motors involved in positioning are absent, a widespread family of ParA/MinD (A/D) ATPases spatially organize plasmids, chromosomes, and an array of protein-based organelles, many of which are fundamental to cell survival and pathogenesis.*” (Line 18)

102, the authors mention that also some pathogens have a set-up with several A/D-ATPases? If this is of particular interest (why should it), these species should be named.

There are many, and the extensive list is available as part of Tables S1 and new Table S2. Instead of highlighting a few pathogenic species, we have added a line highlighting the various genera that have pathogenic species with multiple A/D ATPases: “...we identified several pathogens including *Clostridia*, *Burkholderia*, *Mycobacteria*, *Vibrio*, *Pseudomonas*, and *Xanthomonas* species). We also identified the non-pathogenic and experimentally tractable organism, *H. neapolitanus*...” (Lines 91-93)

120, is FlaG a commonly used abbreviation? I instantly think of flagellar genes (but that maybe only me).

We agree with the reviewer that the abbreviation “FlaGs” can be confusing, especially when used in papers studying the flagella. Saha et al developed and named this bioinformatics tool in 2020 (<https://doi.org/10.1093/bioinformatics/btaa788>). We agree the abbreviated name can be easily confused with the name of flagellar genes and the use of FLAG tags. However, to ensure the developers get proper recognition for their tool, we decided to keep the FlaGs abbreviation in the paper. To avoid confusion, we

added a few sentences to explain FlaGs analysis. A google search of FlaGs analysis also takes the reader to this tool developed by Saha and colleagues.

238, do the authors really mean swarming across a surface? Or spreading in soft agar?

The reviewer is correct that it is spreading in soft agar as opposed to swarming behavior. We have made the clarification as shown: "*Deletion of the A/D ATPase alters chemotaxis cluster number and positioning in cells, which results in a reduction in spreading motility in soft agar.*" (Line 247)

Thank you!

318, 6B, significance for ParA and MinD?

We have added statistical significance measures to all graphs that had them missing. Apologies.

Reviewer #3:

This is a paper full of useful information on the ParA/MinD family of proteins and will serve as an important resource for the scientific community. The work is extensive, well done, well presented and rich in information. The authors do an extensive bioinformatic analysis of this protein family and document the existence of members of this family throughout the bacterial world. I commend the authors for putting together such a complete picture and not trying to publish this piecemeal.

Thank you! It was a tough decision to make, especially pre-tenure. But in the end, it was the right move.

One comment, however, is that the authors set up somewhat of a false premise by wondering about whether "multiple A/D ATPases can coexist and coordinate the positioning of such diverse cargos in the same cell." (lines 10-12 in the abstract). The predictable answer to what is required is summarized nicely between lines 341-4: 1) dimerization interface of the ATPase; 2) interaction with the matrix; and 3) link between ATPase and cargo. I say predictable because of the work already done with MinD, ParA and the authors work on McdA. It seems like the only way these might interfere with each other is if they competed for the same matrix, which is unlikely due to their small number of molecules per cell. Having said that this work establishes the fact that so many family members can readily coexist in the same cell and the bioinformatics and in silico predictions reinforces the principles laid out between lines 341-4 as mentioned above.

We thank the reviewer for their critical assessment of our work. We disagree that the "if" question is a false premise. For example, one source of plasmid incompatibility comes when ParAB systems are too similar to one another. These positioning systems cannot coexist and ensure faithful genetic inheritance. There are also papers showing that ParA ATPases can form heterodimers. In these cases, there is certainly potential co-regulation, cross talk, and/or interference among their respective positioning systems. In a similar fashion, many bacteria encode for different FtsZ copies that can associate and regulate/buffer their activities in Z-ring formation. Finally, it can be imagined that these positioning systems have different expression profiles. Some systems may be similar and show crosstalk if expressed at the same time. Alternatively, it is possible that some positioning systems are turned off at certain points of the cell cycle, while others are turned on. Without formal experimentation, I think it is fair to argue the premise above is not false or misleading.

However, we completely understand and appreciate the reviewers point here. Obviously, these ATPases have evolved to coexist and coordinate their positioning roles in the cell. But the more important question we are trying to answer is not “if”, but “how”. We made modifications to the text to ensure the reader does not think we are working off a false premise.

Other comments:

The extensive tables indicate how many A/D proteins are present in each organism but omit those, if any exist (~5%; line 85), that do not have any. Does that mean that there are convincing chromosome sequences available are there any that have none? Is it 5%? Could they be included in the Table?

We apologize for the omission of important data that we had in-hand. We have added Table S2 to show all bacteria, including those without any A/D ATPases encoded. We also added the caveat that some of these bacteria may still encode A/D ATPases that were not detected either because the bacterial species did not have a fully assembled genome (still in scaffolds or contigs) or because the A/D ATPases diverged to the point where they fell below our threshold for categorizing a protein as part of the A/D ATPase family. The tBLASTn analysis was done by setting the threshold at E.value <0.0001, against the RefSeq Representative genomes database. Under these parameters, roughly 4% (499) of the species (out of 12,569 species) had no sequence homology to ParA-like proteins. However, a similar analysis when done against a nucleotide database (nr/nt) for these selected species, several non-type strain hits were seen. Hence species in the RefSeq Representative genome database, partly due to incomplete genome assembly status, may not accurately identify all species that do not encode A/D ATPases. A supplementary table listing the 4% of species and their respective RefSeq Representative genomes accessions is now available in Table S2.

Or another way to ask it is of the organisms in the Table what fraction have MinD or ParA for example. We already know *E. coli* lacks ParA and there clearly are organisms that lack MinD or others. Could this information be added to the Table? So for example, if an organism has 5 unique ATPases what are they?

We thank the reviewer for their interest in the diversity and spread of A/D ATPases across the bacterial world. To assist the field in identifying the number and type of A/D ATPase(s) in their bacterium of interest, we have added an additional table (Table S2). Here we provide the raw data from our BLAST search, which including the bacterial species, the number of A/D ATPases, their corresponding accession numbers, and predicted type of A/D ATPase (ParA, MinD, etc.). We believe Table S2 will become a valuable resource to the entire microbiology community and its addition as broadened the impact of the paper.

There are several interesting observations about this organism as the authors point out and are the subject of future studies. ParA affects chromosome segregation resulting in anucleate cells but has almost no effect on division. This is clearly quite different than what occurs in *Caulobacter*, for example, where I believe ParA is essential. Also, it may be that bacteria that do multifork replication do not have ParA (*Caulobacter*, *Vibrio*) whereas those that do, do not have ParA (*E. coli*, *Salmonella*, *Bacillus* etc.)

H. neapolitanus and many other bacteria that encode ParABS systems can undergo multifork replication. We agree these are fascinating questions that warrant further future investigation. What necessitates an A/D ATPase for positioning a certain fundamental cellular cargo in one bacterium and not in another remains an open question and will likely uncover key differences in bacterial physiology.

Line 82. I would change ‘can be’ to ‘are’

Corrected.

Line 139. Chromosome segregation in most bacteria. Not sure if this is true. *Caulobacter* and *Vibrio* have ParA but *E. coli* and *B. subtilis* do not (actually *B. subtilis* has Soj but is quite different than ParA).

The statement in question is correct. Multiple bioinformatic analyses have shown that ParABS systems are encoded on the majority of chromosomes (70%) and naturally occurring low-copy plasmids (For example, Livny et al., 2007). What's true for *E. coli* (and *B. subtilis*) may be true for the elephant, but not always true for other bacteria!

Line 280-285 and Fig. 3. Δ FliG-clearly has delayed ParB foci (Fig. 4B) but division does not appear delayed.

We agree with the reviewer that the effect is almost certainly on DNA replication, and not the onset cell division. But without a detailed quantification, we prefer to leave both options open as formal hypotheses that we will address in a future work.

Reviewer #1 (Remarks to the Author):

The manuscript by Pulianmackal and colleagues, entitled "Multiple ParA/MinD ATPases coordinate the positioning of disparate cargos in a bacterial cell" has been improved by the authors following the reviewers concerns and comments. Their responses in the rebuttal are appropriated, and this work is suitable for publication in Nature Communications.

Reviewer #2 (Remarks to the Author):

The authors have fully addressed my concerns (however, please correct HupB/HubP), and I am looking forward to further studies on this great new model system. Fantastic work, congratulations.

Best wishes

Kai Thormann

Reviewer #3 (Remarks to the Author):

The authors have responded very well to the previous round of reviews including doing complementation controls. I only have a few minor comments.

- 1) You point out that this bug has MinD/MinE. What about MinC? My brief look indicates it does. Is it linked? Perhaps it should be mentioned.
- 2) Line 122. Could delete "Conveniently". I would say it is convenient for analysis but expected.
- 3) Line 406. Comment "Not actively positioned". Isn't it likely that it is actively positioned in *E. coli* but not clear how?
- 4) Line 408. Seem to be a limit of the number? Not sure about this. Is there really a limit or only so many needed. I think *Borellia* may have many ParA/ParBs for all the plasmids it has.
- 5) Page 465 ffl. A thought struck me in that having so many ParA/MinD systems might be a liability in that deleting one affects the function of others.

Point-by-Point Response to Reviewer Comments:

Reviewer #1 (Remarks to the Author):

The manuscript by Pulianmackal and colleagues, entitled "Multiple ParA/MinD ATPases coordinate the positioning of disparate cargos in a bacterial cell" has been improved by the authors following the reviewers concerns and comments. Their responses in the rebuttal are appropriated, and this work is suitable for publication in Nature Communications.

Thank you!

Reviewer #2 (Remarks to the Author):

The authors have fully addressed my concerns (however, please correct HupB/HubP), and I am looking forward to further studies on this great new model system. Fantastic work, congratulations. Best wishes, Kai Thormann

Thank you Kai! We are particularly grateful for your comments concerning our work on FlhG and ParC. The additional experiments and revisions have significantly improved the paper. Much appreciated!

HupB has been corrected to HubP. Good catch. Thank you.

Reviewer #3 (Remarks to the Author):

The authors have responded very well to the previous round of reviews including doing complementation controls. I only have a few minor comments.

1) You point out that this bug has MinD/MinE. What about MinC? My brief look indicates it does. Is it linked? Perhaps it should be mentioned.

Yes, *H. neapolitanus* has *minC*. But it is not immediately adjacent to *minD* and *MinE*. This is stated in the legend of Supplementary Figure 1, Line 973: "*minC* in *H. neapolitanus* was found elsewhere in the genome." We also state this in the legend of Supplementary Figure 3, Line 1006 "*minC* is also present elsewhere in the genome."

2) Line 122. Could delete "Conveniently". I would say it is convenient for analysis but expected.

Revised as suggested.

3) Line 406. Comment "Not actively positioned". Isn't it likely that it is actively positioned in *E. coli* but not clear how?

Agreed. The statement now reads "For example, many of the cellular cargos we found here to be positioned by A/D ATPases in certain bacteria, like *H. neapolitanus*, are not actively positioned by A/D ATPases in others, like *E. coli*."

4) Line 408. Seem to be a limit of the number? Not sure about this. Is there really a limit or only so many needed. I think *Borellia* may have many ParA/ParBs for all the plasmids it has.

We agree that the limit may be set simply by the number of cargos to be positioned. But regardless of what is setting the limit, there still seems to be one across the bacterial world. Therefore we've kept the statement.

5) Page 465 ffl. A thought struck me in that having so many ParA/MinD systems might be a liability in that deleting one affects the function of others.

This is an exciting possibility that we've thought about, and we are pursuing!

Thanks again for the thoughtful comments and suggestions that have significantly strengthened the paper. Much appreciated!